# Adaptation to mutational inactivation of an essential gene converges to an accessible suboptimal fitness peak

**João V Rodrigues, Eugene I Shakhnovich***

Department of Chemistry and Chemical Biology, Harvard University, Cambridge, United States

**Abstract** The mechanisms of adaptation to inactivation of essential genes remain unknown. Here we inactivate *E. coli* dihydrofolate reductase (DHFR) by introducing D27G,N,F chromosomal mutations in a key catalytic residue with subsequent adaptation by an automated serial transfer protocol. The partial reversal G27- > C occurred in three evolutionary trajectories. Conversely, in one trajectory for D27G and in all trajectories for D27F,N strains adapted to grow at very low metabolic supplement (folAmix) concentrations but did not escape entirely from supplement auxotrophy. Major global shifts in metabolome and proteome occurred upon DHFR inactivation, which were partially reversed in adapted strains. Loss-of-function mutations in two genes, *thyA* and *deoB*, ensured adaptation to low folAmix by rerouting the 2-Deoxy-D-ribose-phosphate metabolism from glycolysis towards synthesis of dTMP. Multiple evolutionary pathways of adaptation converged to a suboptimal solution due to the high accessibility to loss-of-function mutations that block the path to the highest, yet least accessible, fitness peak.
DOI: https://doi.org/10.7554/eLife.50509.001

**\*For correspondence:**
shakhnovich@chemistry.harvard.edu

**Competing interests:** The authors declare that no competing interests exist.

## Introduction

When important cellular functions are inactivated, for example by genetic mutations, long ranging disruptions of cellular networks can occur, which poses a major adaptive challenge to cells. Recent studies investigated evolution of *E. coli* upon inactivation of non-essential enzymes of carbon metabolism that lead to re-wiring through less efficient pathways (*Krüsemann et al., 2018*; *Long et al., 2018*; *McCloskey et al., 2018a*; *McCloskey et al., 2018b*; *McCloskey et al., 2018c*; *McCloskey et al., 2018*). These studies highlight a crucial interplay between regulatory responses and imbalances in metabolite concentrations resulting from gene knockouts, which can be subsequently corrected by mutations elsewhere. Importantly all these studies involved gene-knockout so that adaptation by reversal to wild type genotype was not possible.

Less is known about the adaptation mechanisms that follow inactivation of unique cellular processes that are deemed indispensable in microbes. The genes that perform such functions, often classified as essential, are believed to face higher selective pressure and thus evolve slower (*Luo et al., 2015*). It is thus expected that functional disruption of essential genes either by mutation or by antibiotic stress can create a major barrier to the adaptability of bacteria. Nevertheless, a comprehensive study involving knock outs of >1000 genes classified as essential in yeast has shown that a small percentage of mutants could recover viability after laboratory evolution (*Liu et al., 2015*), revealing that essentiality can, in some cases, be overcome through adaptation. Loss of essential biosynthetic genes have also been observed to occur naturally, under evolutionary conditions where pathway products are provided externally (*D'Souza and Kost, 2016*), highlighting a context-dependent nature of essentiality. Nevertheless, while adaptation upon inactivation of essential genes in microbes has been demonstrated, its mechanisms remain unknown.

**eLife digest** Predicting how viruses and bacteria evolve remains a challenge. The ability to anticipate when and how bacteria might develop drug resistance would make treating life-threatening diseases easier and could potentially help prevent drug resistance altogether. Studying bacterial evolution under different conditions and cataloguing all possible DNA mutations that allow these bacteria to survive are crucial steps in predicting the appearance of drug resistance.

Studies have revealed that bacteria can adapt to sources of stress, such as antibiotics, in different ways – each involving mutations in distinct genes. However, not all the mutations provide the same benefits to the organism, and the factors that influence how bacteria will adapt are unclear.

Now, Rodrigues and Shakhnovich have used a new approach to push the adaptation ability of the bacterium *Escherichia coli* to the limit. First, they genetically engineered different *E. coli* strains by introducing distinct mutations to an enzyme the bacterium needs to make DNA. These mutations make the resulting strains dependent on external molecules to synthesize new DNA. Next, the cells were grown in conditions where the supply of these DNA precursors was progressively decreased, forcing them to adapt. The obvious way for bacteria to adapt to these conditions would be to 'revert' the mutations that Rodrigues and Shakhnovich introduced in the first place. By using this approach, Rodrigues and Shakhnovich were able to test how often the obvious evolutionary solution happens compared with presumably less-preferred alternative routes.

In rare cases, a specific mutation did restore the activity of the enzyme that enabled DNA synthesis. However, in most cases the bacteria found a different evolutionary solution whereby they all adapt to the decrease in external DNA precursors in the same way, but not by reverting the original mutation. Instead, further mutations disrupt the activity of two metabolic genes, allowing the cells to use the external DNA precursors more efficiently, and keep building DNA. These cells are therefore able to survive even when the levels of the external DNA components are very low, but they will die in the complete absence of these precursor molecules.

This evolutionary solution leads to a non-optimal effect: mutations that restore the activity of the original enzyme, which are the best solution when the two metabolic genes are intact, are no longer as effective. This finding represents a clear example of interactions between genes determining evolutionary outcomes. Rodrigues and Shakhnovich showed that, since it is more likely for a random mutation to disrupt a gene than to revert a previous mutation, adaptations that are less-than-optimal but still work might predominate simply because they happen faster.

Understanding why certain evolutionary adaptations prevail is an important step in predicting evolution and may lead to breakthroughs in many areas. For example, if scientists can identify mutations likely to make bacteria resistant to drugs, it may be possible to act proactively against the bacterial strains that carry those mutations. Eventually, if the factors that lead to specific adaptations are known, it may be possible to exploit this knowledge to create weaknesses in the bacteria's own defences.

DOI: https://doi.org/10.7554/eLife.50509.002

Here we study evolutionary adaptation upon functional inactivation of dihydrofolate reductase (DHFR), an essential *E. coli* enzyme. As the cellular target of the antibiotic trimethoprim, DHFR has been repeatedly observed to accumulate mutations leading to extremely high drug resistance levels in various experimental evolution studies (*Rodrigues et al., 2016*; *Tamer et al., 2019*; *Toprak et al., 2012*). Past efforts to link fitness effects of chromosomal variation in the *folA* locus encoding DHFR and their biophysical effects on DHFR protein allowed us to develop an accurate quantitative bio-physical model of DHFR fitness landscape (*Bershtein et al., 2015a*; *Bershtein et al., 2013*; *Bershtein et al., 2012*; *Bershtein et al., 2015b*; *Rodrigues et al., 2016*). The availability of a clear genotype-phenotype relationship makes DHFR an excellent model to study the dynamics and out-comes of evolution to recover from its inactivation by point mutations. In contrast to previous studies that used gene knockouts, here we inactivate the chromosomal *E. coli* DHFR by introducing muta-tions in the *folA* locus at a key catalytic residue in position 27, generating strains that express inac-tive DHFR (with at least 4 orders of magnitude lower catalytic efficiency than the wild type). These strains are viable only with external metabolic compensation, allowing the mutants to adapt to lack

of DHFR function by decreasing supplement concentration. The advantage of this approach is that it presents cells with an obvious evolutionary 'solution' of reverting the mutant back to wild type variant without massive rewiring that could lead to potentially lesser fit variants. However, an actual outcome may depend on evolutionary dynamics which could revert to wild type or other form of active DHFR (higher fitness peak) or converge to a potentially more accessible solution of rewiring to a less efficient metabolic pathways that do not require DHFR function. Therefore, this setup allows us to assess relative roles of height and accessibility of fitness peaks in determining the outcome of evolutionary dynamics.

We show that, depending on starting DHFR variant, partial reversion of DHFR phenotype may indeed occur. However, adaptation to low concentration of external metabolites through metabolic rewiring is the prevalent evolutionary solution due to the availability of a greater number of trajectories leading to consecutive gene inactivation events in two key loci, *thyA* and *deoB*. Using omics analysis, we observe global perturbations in metabolites and proteins levels in both naïve D27 mutant strains and in evolved strains. Finally, we show how adaptation to loss-of-function mutations in DHFR, the cellular target of the antibiotic trimethoprim, can lead to high levels of drug resistance.

## Results

DHFR is encoded by the *folA* gene and is essential for the biosynthesis of purines, dTTP, glycine and methionine. We sought to inactivate the *folA* gene product in *Escherichia coli* by introducing mutations in the key catalytic residue D27 (*Figure 1A*).

To construct these strains we supplemented the growth medium with a mixture (folAmix) of adenine, inosine, thymine (precursor of dTTP), methionine and glycine which allows cells to overcome DHFR deficiency (*Howell et al., 1988*; *Kwon et al., 2010*). We envisaged that inactivated DHFR mutants growing in the presence of supplement folAmix could be progressively challenged to adapt to ever decreasing concentrations of this supplement in the growth medium (*Figure 1B*). Next, we implemented a fully automated serial passaging scheme as depicted in *Figure 1C–D* using a Tecan liquid handling robot. In this setup, the growth rate of replicate cultures is monitored by periodic OD readings, and the concentration of folAmix is adjusted downward in each serial passage when cultures exceed a defined growth rate threshold. This feedback control loop ensures mutant strains are continuously challenged to grow at sub-optimal conditions, sustaining a strong selective pressure on the loss of DHFR function. This approach combines the medium-throughput capabilities of plate-based serial dilution methods (currently, up to 48 independent replicate trajectories can be evolved in parallel), and both real-time monitoring of fitness status and control of growth conditions featured in a morbidostat setup (*Toprak et al., 2012*; *Toprak et al., 2013*), which allows cells to continuously experience exponential growth and sustained selection pressure.

### D27 mutations confer growth defects

We selected single or double nucleotide base mutations to replace a key catalytic residue Asp 27 either for Asn, Gly or Phe. The presence of a carboxylate side chain in position 27 is strictly conserved among 290 bacterial DHFR orthologues, where Asp and Glu are present at a frequency of 82% and 18%, respectively (*Figure 2—figure supplement 1A*). These replacements were chosen to explore how different degrees of structural perturbations could potentially lead to alternative mutational pathways; while asparagine is structurally similar to aspartate, the introduction of residues with side chains that are either bulky and hydrophobic (phenylalanine) or absent (glycine) is expected to create significantly more structural perturbations, at least locally at the active site.

Purified D27 mutant proteins were characterized (*Table 1*). The catalytic activity measurements confirm the lack of significant DHFR function in these variants; catalytic efficiencies ($k_{cat}/K_M$) are several orders of magnitude lower than wild type. However, thermal denaturation data obtained for the D27 mutant proteins indicates that their stability is mostly unaffected (or even significantly increased in the case of D27F mutant; *Tian et al., 2015*) showing that, despite the lack of catalytic activity, these proteins retain the ability to fully fold. Importantly, this provides a pathway for restoration of DHFR function over the course of evolution, either by revertant mutations at the D27 locus or, potentially, compensatory mutations elsewhere in the protein. D27 mutant strains were generated by lambda red recombination (*Bershtein et al., 2012*) and plated in folAmix-containing agar media, however, growth was only visible after 48 hr and the colonies formed were miniscule. When DHFR

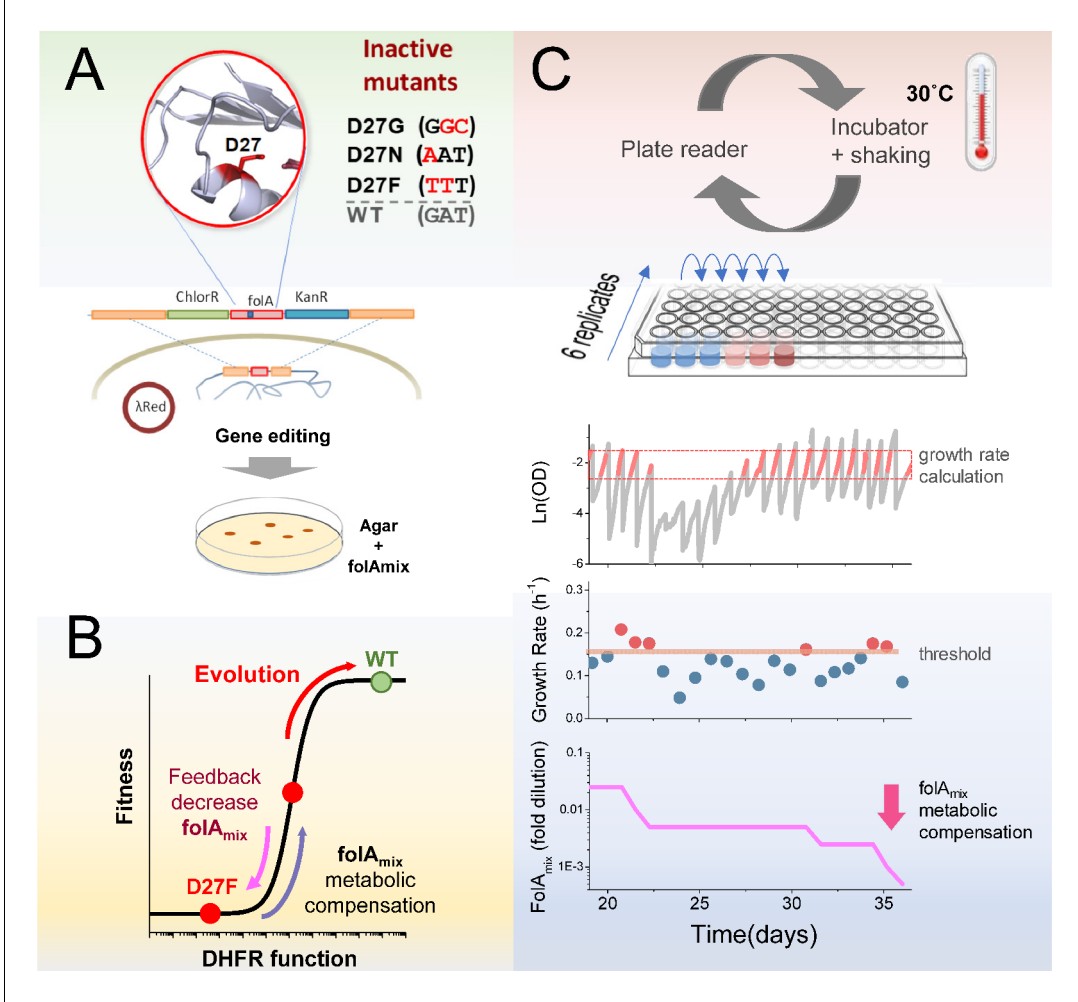

**Figure 1.** Automated experimental evolution of inactive DHFR mutants. (**A**) Mutations in key D27 catalytic residue of DHFR were introduced in *E. coli* chromosome, together with flanking antibiotic resistant markers, by lambda red recombination and strains devoid of DHFR function were selected with antibiotics in folAmix-containing agar plates. (**B**) D27 mutants are lethal but fitness can be partially rescued by metabolic compensation with folAmix. Adaptative changes that increase fitness can be counterbalanced by decreasing the concentration of folAmix in the growth medium, forcing the cells to evolve without DHFR function. (**C**) Experimental evolution using automated liquid handling. Six replicates of cell cultures are placed in the first column of a 96-well plate and incubated at 30 °C with shaking and the optical density is measured every 30 min. The cultures are grown until the average OD reaches a threshold (0.3), that was defined to avoid nutrient limitation and consequent shift into the stationary phase. At this point cultures are diluted simultaneously into the wells of the adjacent column, to a common starting OD (0.01), and the cycle is repeated. The growth rate is calculated for every culture and, whenever it exceeds a defined threshold, the amount of folAmix is independently decreased for each culture in the subsequent dilution.
DOI: https://doi.org/10.7554/eLife.50509.003

function was assayed in cell lysates of D27 mutant strains, no DHFR activity could be detected (*Figure 2—figure supplement 1B–C*), confirming that these mutations inactivate DHFR function in the cell. As expected, growth experiments showed that D27 mutants grow only at high concentrations of folAmix (*Figure 2A*).

Interestingly, D27N and D27G mutants show slightly better growth at lower concentrations of folAmix than D27F. Given that the catalytic efficiencies of D27G and N, albeit extremely low, are 2 orders of magnitude higher than that of D27F, it is possible that such difference could have a beneficial impact at low folAmix concentrations. In addition, potential acquisition of a slightly advantageous mutation elsewhere upon genetic manipulation to introduce D27G/N mutation in the *folA* locus can also lead to growth differences (see Discussion). We tested D27 mutants for growth in the presence of individual components of folAmix and their different combinations and found that only thymine was essential, although growth with thymine alone is slower compared to growth with all

**Table 1.** In vitro properties of DHFR mutant proteins [a]

|  | Wild type | D27F | D27N | D27G | D27C |
|---|---|---|---|---|---|
| $k_{cat}$ (s$^{-1}$) | 13 ± 2 | (5 ± 1)×10$^{-4}$ | (2.6 ± 0.9) × 10$^{-2}$ | (1.8 ± 0.4)×10$^{-2}$ | 3 ± 0.8 |
| $K_M$ (µM) | 0.8 ± 0.2 | (1.2 ± 0.6)×10$^2$ | (4 ± 1)×10$^1$ | 23 ± 7 | (7 ± 2)×10$^1$ |
| $k_{cat}/K_M$ (s$^{-1}$ µM$^{-1}$) | 16 ± 1 | (4 ± 1)×10$^{-6}$ | (7 ± 0.8)×10$^{-4}$ | (8 ± 2)×10$^{-4}$ | (4.2 ± 0.8)×10$^{-2}$ |
| $k_{cat}/K_M$ relative to wild type | - | 2 × 10$^{-7}$ | 4 × 10$^{-5}$ | 3 × 10$^{-5}$ | 2 × 10$^{-3}$ |
| $K_i$ (nM) | 1.0 ± 0.3 | ND | ND | ND | (1.7 ± 0.7)×10$^4$ |
| $\Delta T_m$ ($^0$C) | - | +7.6 ± 0.1 [b] | +1.1 ± 0.3 | +1.2 ± 0.2 | +0.6 ± 0.6 |
| bis-ANS | 1 | 1.1 ± 0.1 | 2.9 ± 0.3 | 1.7 ± 0.3 | 2.2 ± 0.5 |

[a] Kinetic properties for dihydrofolate reductase catalytic activity ($k_{cat}$ = enzymatic turnover number, $K_M$ = Michaelis Menten constant, $k_{cat}/K_M$ = catalytic efficiency, $K_i$ = inhibition constant for trimethoprim) and protein stability properties ($\Delta T_m$ = difference in melting temperature of folding with respect to wild type, bis-ANS = relative fluorescence from binding of bis-ANS to molten-globule intermediates with respect to wild type protein).

DOI: https://doi.org/10.7554/eLife.50509.010

folAmix components (*Figure 2—figure supplement 2*). The growth rates measured for all D27 mutants at high folAmix fall in the range 70–80% of wild type and lag times were typically 1.5–2 fold longer (*Figure 2B*).

## D27 mutations cause major metabolic and proteomic changes

The previous observation that D27 mutant strains show severe growth defects suggests that DHFR inactivation imposes considerable homeostatic imbalance. We focused on the strain D27F to carry out a detailed characterization of the systems-level effects of DHFR inactivation. To that end we carried out high throughput proteomics analysis of D27F mutant strains using LC/MS TMT approach as described earlier (*Bershtein et al., 2015a*). The method based on differential labeling provides abundances of proteins in the proteome relative to a reference strain which in our case was wild type (*Bershtein et al., 2015a*). We computed Z-scores of the log of relative (to wild type as reference) abundance according to the following equation:

$$z_i^{strain/ref} = \frac{Y_i^{strain/ref} - \left\langle Y^{strain/ref} \right\rangle}{\sigma_Y^{strain/ref}}, \tag{1}$$

where index $i$ refers to protein, $Y_i^{strain/ref} = Log_{10}\left(\frac{A_i^{strain}}{A_i^{ref}}\right)$ where $A_i^{strain}$ and $A_i^{ref}$ are the protein i abundances obtained for the mutant and reference (wild type) strains, respectively, $\left\langle Y^{strain/ref} \right\rangle$ denotes a quantity $Y_i^{strain/ref}$ averaged over all proteins for a given strain, and $\sigma_Y^{strain/ref}$ is the standard deviation of $Y^{strain/ref}$ (see *Supplementary file 1*). Then, we classified the proteins by their cellular function according to *Sangurdekar et al. (2011)* and performed a Student t-test to determine which groups of proteins had statistically significant variation of protein levels in D27F strain, with respect to wild type (see *Supplementary file 2*). Numerous cellular processes were significantly altered, reflecting broad genome-wide effects of DHFR inactivation (*Figure 2—figure supplement 3*). Proteins involved in central processes were significantly downregulated in D27F strain, including energy metabolism (aerobic respiration, TCA cycle), metabolism of nitrogen, several amino acids, pyrimidines and lipopolysaccharides. On the other end, we found significant increase in the expression of proteins involved in stress responses, peptidoglycan recycling and salvage of guanine and xanthine. We then performed both targeted and untargeted metabolomic analysis of D27F mutant strain and wild type (see experimental details) to characterize significant changes at the level of metabolites. Likewise, Z-scores for metabolite levels were computed for D27F mutant, with respect to wild type (see *Supplementary file 3*). The metabolites with the highest absolute Z-scores (>1.96) were selected for an enrichment test using MBRole online software (*López-Ibáñez et al., 2016*) to identify pathways in which altered metabolites are overrepresented. The analysis revealed that the metabolism of purines, pyrimidines, beta-alanine, histidine and sulfur were the most significantly changed in D27F mutant comparatively to wild type (*Figure 2—figure supplement 4A–B*). A detailed scheme representing the changes in metabolites and proteins levels of nucleotide synthesis pathways is

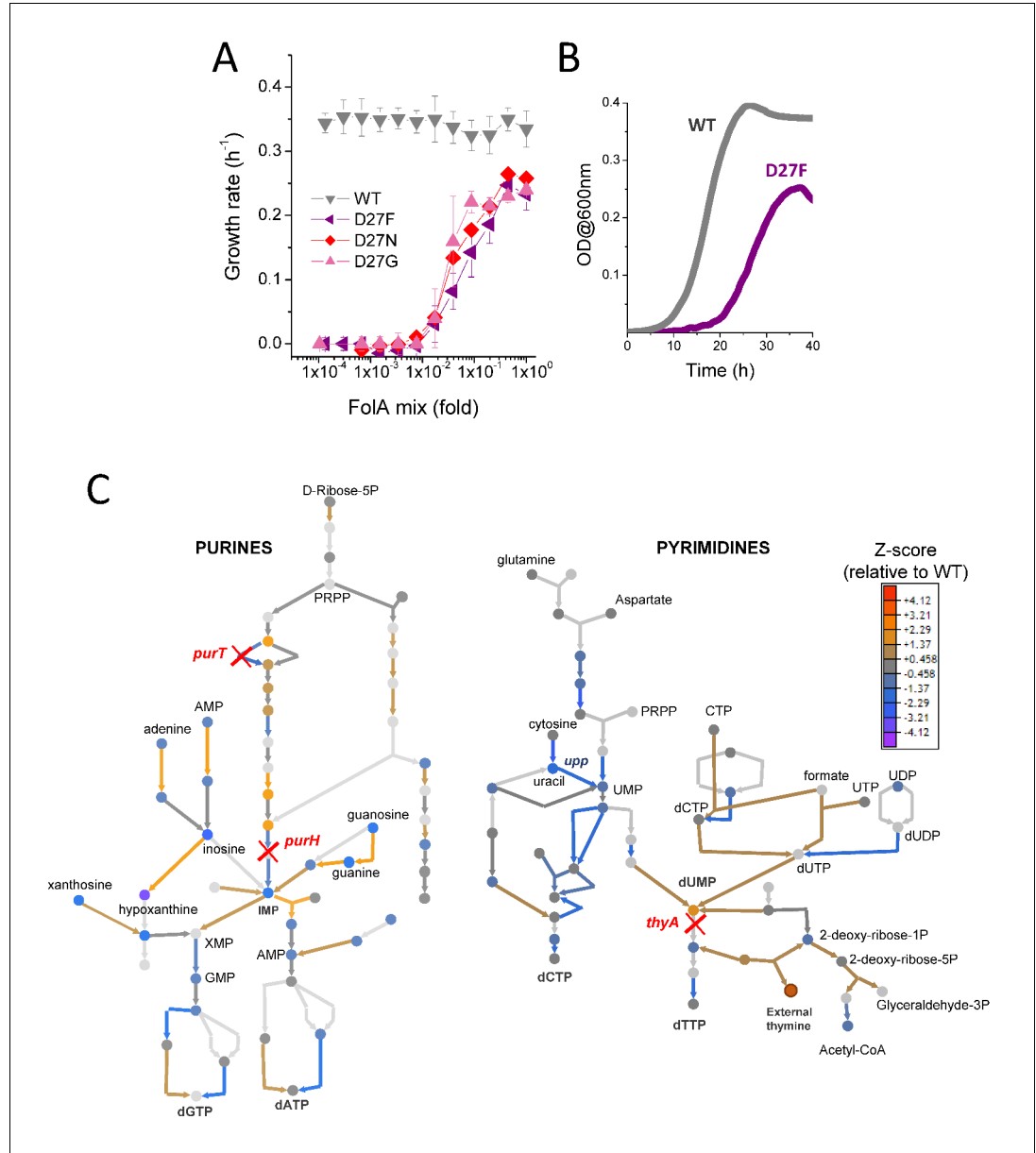

**Figure 2.** D27 mutants require folAmix to grow. (**A**) The growth rates of each D27 mutant and wild type were measured at various dilutions of folAmix, with respect to initial composition (adenine 20 μg/mL, inosine 80 μg/mL, thymine 200 μg/mL, methionine 20 μg/mL and glycine 20 μg/mL), (see also *Figure 2—figure supplement 1* and *Figure 2—figure supplement 2*). Cultures were grown in M9-minimal medium at 30 °C, and absorbance was monitored at 600 nm. Data are represented as mean ± SD (N = 4) (**B**) Comparison of representative growth curves of wild type and D27F mutant obtained in the presence of 1x folAmix. (**C**) Schematic representation of purine and pyrimidine biosynthetic pathways depicting changes in the levels of metabolites and proteins for the D27F mutant (Z-scores, relative to wild type). Metabolites are represented by circles and arrows represent enzymatic reactions color-coded by the levels of associated proteins (see also *Figure 2—figure supplement 3* and *Figure 2—figure supplement 4*). Light gray shading indicates that no data is available. Metabolites that are downstream of the enzymatic reactions requiring reduced folates (ThyA, PurH and PurT) are strongly depleted in D27F mutant, whereas those metabolites immediately upstream to those reactions are increased in respect to wild type.
DOI: https://doi.org/10.7554/eLife.50509.004

The following source data and figure supplements are available for figure 2:

**Source data 1.** D27 mutants require folAmix to grow.
DOI: https://doi.org/10.7554/eLife.50509.009

**Figure supplement 1.** Mutations in key catalytic residue D27 impair DHFR function.

*Figure 2 continued on next page*

*Figure 2 continued*

DOI: https://doi.org/10.7554/eLife.50509.005

**Figure supplement 2.** Naïve D27F strain requires thymine for growth.

DOI: https://doi.org/10.7554/eLife.50509.006

**Figure supplement 3.** D27F mutation causes major proteomic changes.

DOI: https://doi.org/10.7554/eLife.50509.007

**Figure supplement 4.** The abundance of various metabolites is strongly affected in naïve and evolved D27F strains.

DOI: https://doi.org/10.7554/eLife.50509.008

shown in *Figure 2C*. Not surprisingly, we observe build-up of metabolites upstream the reactions that require reduced folate cofactors (PurT, PurH, and ThyA), whereas metabolites downstream of those reactions are found to be strongly depleted.

Overall, weaker growth and small colony phenotype show that D27 mutants are severely compromised even in presence of folAmix. Using the evolutionary scheme described previously (*Figure 1C*), we allowed six replicates derived from the same colony of each mutant strain D27F/N/G to evolve in parallel for about 50 days.

## Evolution of D27G mutant

Cultures of D27G mutant strains were first grown in the presence of folAmix diluted to 0.1 fold with respect to the initially defined folAmix composition.

Throughout the course of the experiment the concentration of supplement mixture further decreased (*Figure 3A*) in response to increasing fitness of the mutant strains, as imposed by the feed-back control loop discussed previously. Thus, the change in folAmix concentration over time reflects the effect of adaptation. Three trajectories (1,3 and 6) stand out by reaching the ability to grow in the complete absence of folAmix. Trajectories 2 and 5 reached a point where growth dropped below detection limit, and ultimately could not be recovered, whereas trajectory four had adapted to grow at low concentrations of folAmix. We hypothesized that mutations in *folA* locus could have restored DHFR catalytic activity in the cultures where reversion of phenotype was observed. Accordingly, Sanger sequencing analysis of this region revealed that in all these three trajectories (1,3 and 6), but not trajectory 4, residue Gly27 had mutated to cysteine. This finding was surprising because alignment of multiple known DHFR sequences shows that cysteine does not occur naturally in position 27; only aspartate or glutamate are observed in this locus (*Figure 2—figure supplement 1A*). To assess whether Cys27 mutant is functional, this variant was purified and characterized in vitro, and its catalytic properties are compared with wild type and other mutants in *Table 1*. Although stability-wise D27C mutant is similar to wild type protein, it shows much weaker catalytic efficiency, mostly in terms of high $K_M$, retaining only about 0.1% $k_{cat}/K_M$ of wild type. To assess if this residual catalytic function is sufficient to explain the reversion of phenotype in the evolved strain we first predicted fitness based on a previously established model (*Rodrigues et al., 2016*). Taking as input the in vitro biophysical properties of purified D27C mutant, namely $k_{cat}/K_M$ and bis-ANS fluorescence values, the model predicts a growth rate of about 30% of the wild type strain. To check the validity of this prediction, the mutation D27C was reconstituted in the wild type background by lambda red recombination and cells were plated in folAmix-containing agar plates to lift any selective pressure for DHFR activity. This strain was able to grow in the absence of folAmix supplementation, and the measured growth rate was 92% of the wild type, which is in fair agreement with the in vitro-based prediction which does not take into account a possibility of upregulation in response to DHFR deficiency (*Bershtein et al., 2015a*) (see Material and methods section Predicting fitness of D27 mutants for discussion) (*Figure 3B–D*). This analysis shows that D27C mutation alone explains a significant fraction of fitness recovery of the evolved strains. Of note is also the fact that the significant loss in catalytic efficiency in D27C mutant is associated with a dramatic 4 orders of magnitude increase in the inhibition constant for trimethoprim, and consequent nearly 100-fold increased resistance of the D27C strain to trimethoprim inhibition in comparison with the wild type (*Figure 3E*). This is in line with previous observations that active site mutations compromise catalytic function but also disrupt pocket interactions with the drug, resulting in high levels of resistance (*Bader et al., 2006*; *Rodrigues et al., 2016*).

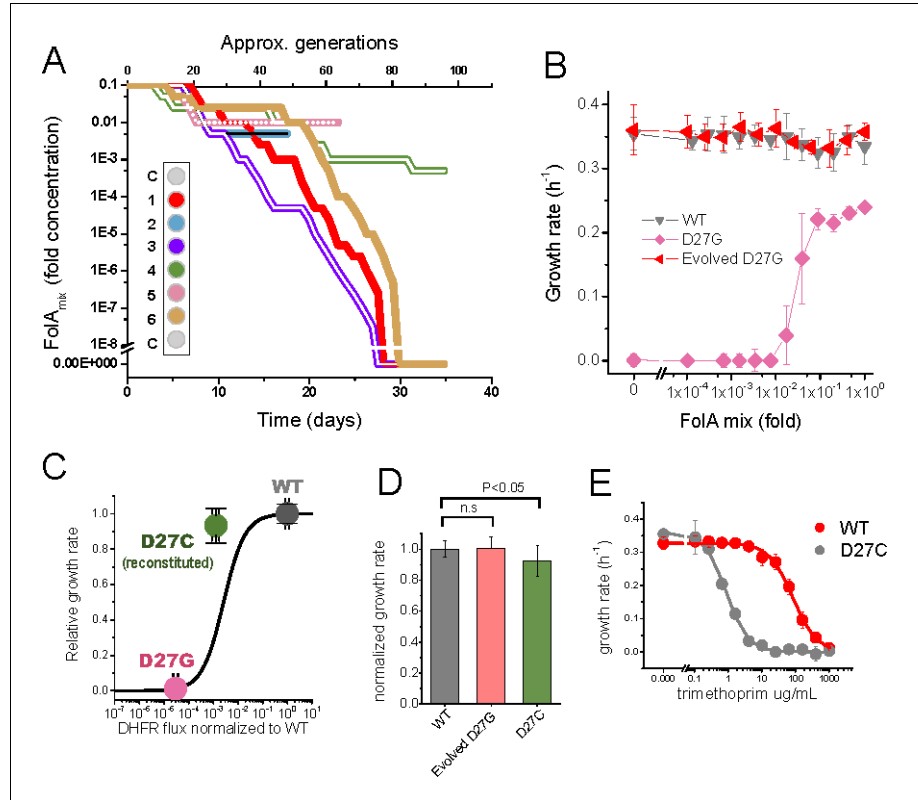

**Figure 3.** Phenotype-reverting G27C mutation emerges upon evolution of D27G strain. (**A**) Evolutionary profiles of the six trajectories in D27G adaptation showing the time dependent changes in folAmix concentration necessary to sustain growth. Each trajectory is colored according to its position in the column of the 96-well plate, as represented in the scheme (C = wells with growth medium only). (**B**) Evolved D27G strain does not require folAmix to grow. The growth rates determined at different concentrations of folAmix are compared for wild type, naïve and evolved D27G strain from trajectory 6 (clone G33T6#1). Data are represented as mean ± SD (N = 4). (**C**) Fitness model (*Rodrigues et al., 2016*) based on the biophysical properties of DHFR mutants (represented by the black line) allows to predict the growth rate of mutant strains from in vitro properties of the purified D27G and D27C DHFR mutant proteins. The dots represent the growth rate determined experimentally for each mutant strain at the flux value predicted from the biophysical properties. The model predicts a growth rate for D27C that is approximately 30% of the wild type value. This is in fair agreement with the experimental value measured for a BW25113 strain in which D27C mutation was introduced in the wild type folA gene (D27C reconstituted) which was 92% of the wild type. (**D**) Comparison of growth rates of wild type, evolved D27G and reconstituted D27C strain (mean ± SD, N = 4). (**E**) Dose-dependent growth inhibition by trimethoprim determined for wild type and reconstituted D27C mutant (mean ± SD, N = 3). The data was fitted with a logistic equation (solid lines) from which IC50 was determined to be 0.8 ± 0.1 μg/mL for wild type and 79 ± 17 μg/mL for D27C mutant.
DOI: https://doi.org/10.7554/eLife.50509.011

The following source data is available for figure 3:

**Source data 1.** Phenotype-reverting G27C mutation emerges upon evolution of D27G strain.
DOI: https://doi.org/10.7554/eLife.50509.012

## Evolution of D27F and D27N mutants

Evolutionary trajectories of D27F and D27N mutants represented in *Figure 4A and B*, respectively, show a marked decrease in folAmix necessary to sustain growth, in most cases reaching concentrations nearly three orders of magnitude lower than initially required for naïve strains.

To verify that these strains adapted to grow at low folAmix concentrations we first plated evolved cultures (from −80 °C glycerol stocks) in agar medium supplemented with 1x concentration of folAmix and then randomly selected individual colonies from different trajectories to measure growth at various concentrations of folAmix. Plating the cultures at a high folAmix concentration ensures that

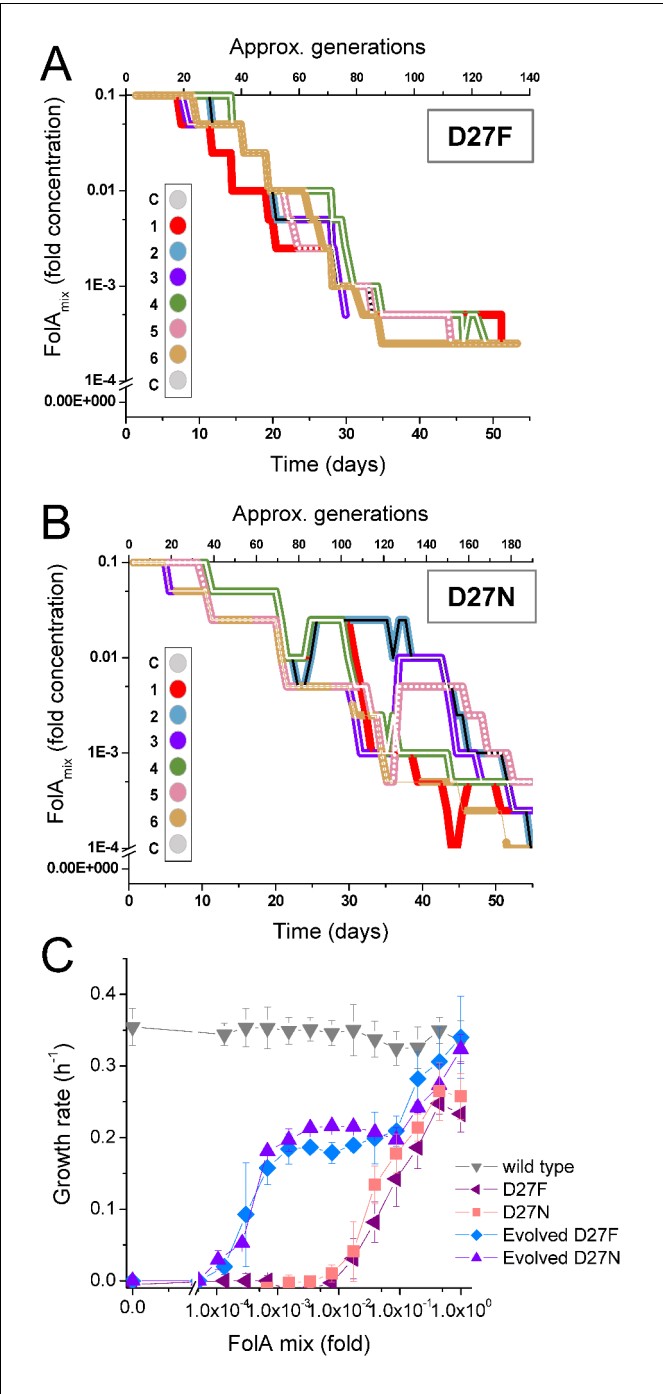

**Figure 4.** D27F and D27N mutant evolved to grow at low folAmix concentrations. Evolutionary profiles of each trajectory in (**A**) D27F and (**B**) D27N adaptation to loss of DHFR function. (**C**) Comparison of growth dependence on folAmix concentration determined for wild type, both naïve and evolved D27F (trajectory 1, clone F51T1#1) strains and naïve and evolved D27N (trajectory 6, cloneN51T6#1), see also *Figure 4—figure supplement 1*, *Figure 4—figure supplement 2*. Data are represented as mean ± SD (N = 4).
DOI: https://doi.org/10.7554/eLife.50509.013

The following source data and figure supplements are available for figure 4:

**Source data 1.** D27F and D27N mutant evolved to grow at low folAmix concentrations.
DOI: https://doi.org/10.7554/eLife.50509.016

**Figure supplement 1.** Evolved D27F and D27N strains require folAmix for growth.
DOI: https://doi.org/10.7554/eLife.50509.014

*Figure 4 continued on next page*

*Figure 4 continued*

**Figure supplement 2.** D27 mutant strains are extremely resistant to trimethoprim inhibition.
DOI: https://doi.org/10.7554/eLife.50509.015

there is no biased selection for high-fitness phenotypes when colonies are randomly picked from agar media. We found that individual colonies taken among all trajectories that adapted to low folA-mix concentrations all showed a strikingly similar phenotype - folAmix-dependent growth profile, irrespective of the initial variant at D27 locus (F/N and also trajectory four in D27G) (*Figure 4—figure supplement 1A–C*). *Figure 4C* shows representative growth profiles obtained with colonies taken from trajectories 1 and 6 of D27F and D27N, respectively, showing a marked decrease in growth rate at concentrations of folAmix comparable to the lowest concentrations achieved during the evolution experiment. However, these strains cannot grow in the complete absence of folAmix. This result indicates that, unlike some trajectories in D27G evolution, adaptation of D27N and D27F strains did not involve reversion of the *folA⁻* phenotype. Accordingly, no mutations were found in *folA* locus in evolved D27N and D27F strains (see *Supplementary file 4*). Other mechanisms must therefore be responsible for the ability to grow at low folAmix concentrations. The growth rate of the evolved strains measured at high concentrations of folAmix reached values comparable to wild type, showing a clear fitness increase with respect to naïve strain. However, at lower concentrations of folAmix the growth rate of evolved strain becomes markedly reduced to nearly half of the wild type value and appears to plateau in an intermediate range of folAmix concentrations. The growth rate value at this plateau coincides with that measured for cells grown in thymine alone (*Figure 4—figure supplement 1D*), which is indicative that thymine is the growth-limiting component at these concentrations of folAmix and that cells utilize thymine less efficiently in the absence of the other folAmix components. We also evaluated sensitivity to trimethoprim for both naïve and evolved D27F strains. Not surprisingly, in the absence of a functional DHFR and in the presence of folAmix D27F mutant strains are extremely resistant to trimethoprim (*Figure 4—figure supplement 2A*). We noted, however, that while no decrease in growth was evident in naïve D27F strain up to 1000 µg/mL, the growth rate of the evolved strain dropped abruptly above 500 µg/mL trimethoprim, suggesting that the drug is acting on an unknown target in the cell which is essential in the evolved but not the naïve strain. We then tested how the resistance of evolved D27F strain would compare with wild type at very low concentrations folAmix (*Figure 4—figure supplement 2B*). We found that in those conditions IC50 for the wild type is similar to that measured in the absence of supplement, yet the evolved D27F strain still shows an extremely high resistance to trimethoprim (IC50 = 656 ± 78 µg/mL).

## Loss of function mutations in thyA and deoB lead to adaptation to low folAmix concentrations

Whole genome sequence (WGS) analysis was performed to identify the genetic basis for the adaptation mechanisms of the evolved strains. We analyzed naïve D27F/N/G strains and individual clones representative of adaptation to low folAmix (evolved D27F - trajectories 1 and 2 and evolved D27N - trajectory 6), and one clone representative of adaptation through DHFR reversion (D27G- trajectory 6). In addition, to characterize the dynamics of mutation fixation, individual colonies from intermediate evolutionary time points of D27F trajectory two were also sequenced. *Figure 5A–B* summarizes the WGS results obtained for each clone of evolved D27 and naïve strains.

We found that D27N and D27G strains, but not D27F, had additional background mutations that must have been acquired prior to starting the evolution experiments; although all D27 strains were constructed from the same wild type *E. coli* parent strain, we could not prevent the appearance of mutations at any given stage of genetic manipulation prior to the evolution experiments. Strain D27N had a point mutation (P74S) in the gene *nanM* that encodes N-acetylneuraminate mutarotase, which supports efficient growth form sialic acid as carbon source (*Severi et al., 2008*), and a point mutation (L44Q) in the gene *thyA* (discussed below). Strain D27G had a point mutation (A101V) in the gene *ymfD* that encodes a putative SAM-dependent methyltransferase and an early stop codon in the gene *upp*, that encodes the enzyme uracil phosphoribosyltransferase that participates in pyrimidine salvage pathway. It is possible that these mutations may provide some fitness advantage

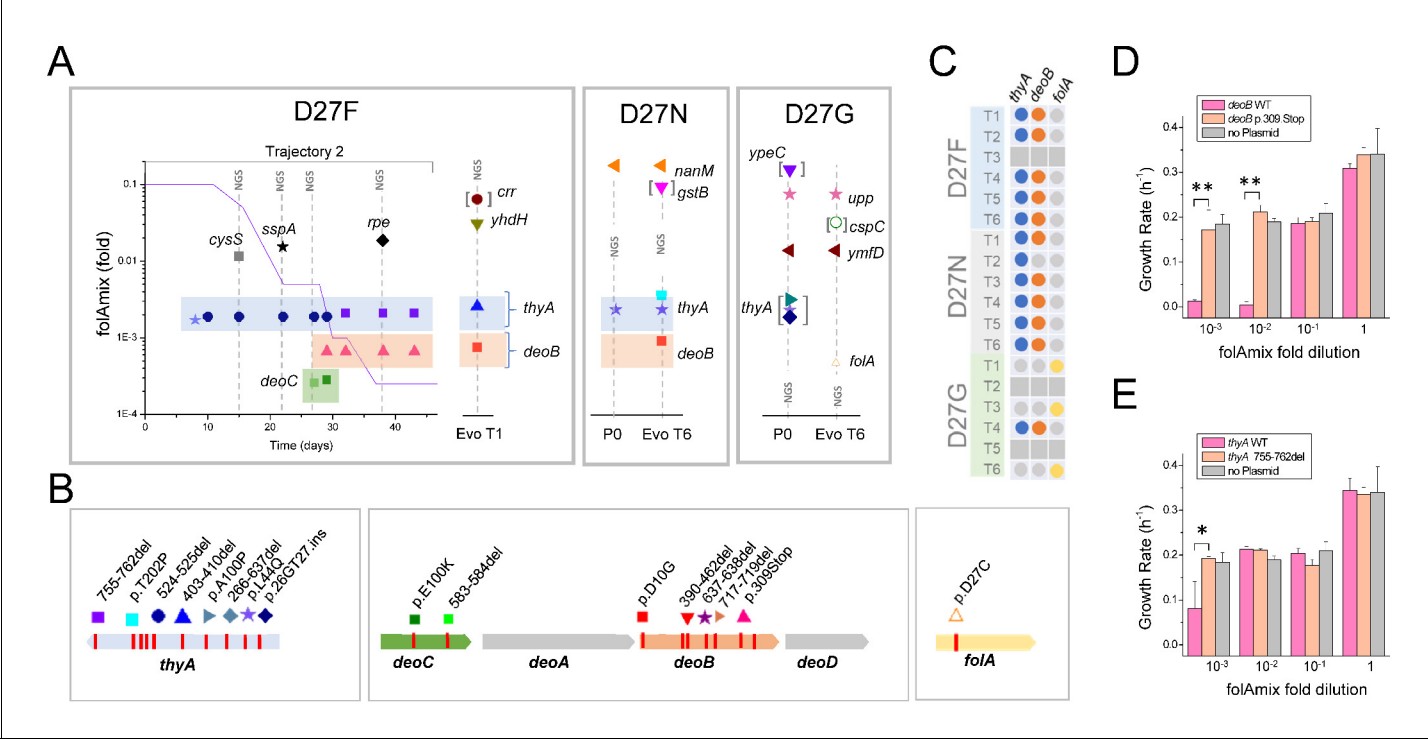

**Figure 5.** Loss of function mutations in two genes, thyA and deoB, lead to adaptation to low folAmix concentrations. (**A**) Mutations identified in naïve and evolved D27 mutant strains. Vertical lines identified with NGS represent mutations identified by whole genome sequencing whereas the remaining cases were identified by Sanger sequencing. Details are presented for trajectory 2 of D27F evolution and include the mutations identified at various passages and the folAmix profile obtained for that trajectory. Points denoted P0 correspond to naïve strains. Mutations between brackets represent polymorphisms with estimated frequencies of 0.5. See also *Figure 5—figure supplement 1* for the effect of upp and ymfD mutation in naïve D27G strain. Figure (**B**) Mutations identified in the most relevant genes. **C**) Mapping thyA, deoB and folA mutations found among evolved strains from all trajectories of each D27 mutant. Colored dots represent the presence of mutations, whereas gray dots represent genes that were not mutated in evolved strains. Trajectories that ceased growth prior to the end of the experiment were not sequenced (gray squares). **D–E**) Expression of wild type DeoB and ThyA in evolved D27F strain reverts phenotype to high folAmix requirement. Evolved D27F strain (trajectory 1, clone F51T1#1), with background mutations deoB p.D10G and thyA 403-410del, was transformed with pTRC-tetR plasmid coding (**D**) wild type or mutant deoB and (**E**) wild type or mutant thyA, under the control of tetR repressor. Evolved D27F strain without plasmid is also represented as control. Cells were grown in M9 minimal medium and growth rates were measured from periodic OD measurements.. *p<0.01, **p<0.001.
DOI: https://doi.org/10.7554/eLife.50509.017

The following source data and figure supplement are available for figure 5:

**Source data 1.** Expression of wild typeDeoB and ThyA in evolved D27F strain reverts phenotype to high folAmix requirement.
DOI: https://doi.org/10.7554/eLife.50509.019

**Figure supplement 1.** Mutations in *upp* and *ymfD* do not impact fitness of D27G.
DOI: https://doi.org/10.7554/eLife.50509.018

with respect to 'pure' D27F strain (see below and discussion). Comparison between evolved and naïve D27G strains reveals that no other mutation was fixed upon evolution besides the one nucleotide change at the D27 locus of the *folA* gene, corresponding to the Gly->Cys substitution described earlier. This is fully consistent with the earlier conclusion that D27C mutation alone explains the fitness recovery observed in D27G evolution. We asked if the presence of mutations in *upp* and *ymfD* genes originally found in D27G strain could have an impact on fitness of the cells, and thus could have conditioned the evolutionary fate of this strain. We note the potential role of the mutation in *upp* gene which protein product levels appeared depleted in proteomics analysis of naïve D27F strain. To address this, we engineered additional D27G strains and, after confirming the absence of mutations in *thyA*, *upp* and *ymfD*, compared their growth properties at various folAmix concentrations with D27G *upp/ymfD* strain that was used in the evolution experiments. No difference was found in the growth rate profiles of these strains (*Figure 5—figure supplement 1*), indicating no

obvious impact of mutations in *upp* and *ymfD* in favoring the reversion of DHFR function over *thyA* inactivation.

From the analysis of evolved D27F and D27N we can identify two common hotspots for mutations, the genes *thyA* and *deoB*, encoding thymidylate synthase and phosphopentomutase, which are involved in the synthesis of dTMP via de novo and salvage pathways, respectively. For that reason, the sequence of these two loci were determined by Sanger analysis for all other trajectories that evolved to grow at low folAmix concentrations, including trajectory four in D27G evolution. Strikingly, in a total of 12 trajectories all were found to have mutations in *thyA* and 11 had *deoB* mutated (*Figure 5C*). We noted as well that evolved D27F strains also carried mutations in other genes, *crr* (7 bp deletion), *rpe* and *yhdH* (4 bp deletion), however, these were not observed among other trajectories indicating that these are most likely either random passenger mutations or provide only marginal advantage on a specific genetic background. On the other hand, the occurrence of deletions and deleterious point mutations (see *Supplementary file 5*) in both *thyA* and *deoB* strongly implies that functional inactivation of these gene products arises as a main mechanism of adaptation to the lack of DHFR function. We reasoned that complementation with either thymidylate synthase or phosphopentomutase should create a significant fitness disadvantage to the *evolved* cells, which can be reverted if these enzymes become inactivated. To verify this prediction, we transformed evolved D27F strains with plasmids expressing either wild type ThyA or DeoB proteins and compared the ensuing growth rates of these strains with transformants carrying the same plasmids expressing functionally inactive ThyA and DeoB, respectively. Expression of functional ThyA and especially DeoB were found to be highly deleterious in *evolved* D27F strains at low folAmix concentrations, as shown *Figure 5 D-E*, whereas no significant change in growth was observed upon expressing inactive mutants.

Overall, these results show two competing adaptation mechanisms to the loss of DHFR activity, either involving a reverting mutation at the D27 locus or, more prevalently, consecutive inactivation of two genes.

## Loss-of-function mutations in deoB prevent draining of deoxyribose pools into energy production

The gene *deoB* is part of an operon responsible for deoxyribose degradation, that includes *deoA*, *deoC*, and *deoD*, which is under the control of 2-deoxyribose-5-phosphate-inducible *deoR* repressor (*Hammer-Jespersen and Munch-Ptersen, 1975*). DeoB catalyzes the interconversion of 2-Deoxy-D-ribose-1-phosphate and 2-Deoxy-D-ribose-5-phosphate, and, while the former is necessary for synthesis of dTMP from thymine by DeoA, the latter can be further degraded in glycolysis, through conversion into acetaldehyde and D-glyceraldehyde 3-phosphate by DeoC (*Figure 6A*).

Therefore, inactivating *deoB* or *deoC* is an 'economy' solution preventing key metabolite in thymine uptake to be wasted in energy production, allowing cells to efficiently use small amounts of thymine available from media supplement for nucleotide synthesis (*Dale and Greenberg, 1972*). In agreement with the key role of deoxyribose in limiting thymine uptake, while *deoB*$^+$ cells cannot grow at low concentrations of thymine in the media, replacing this precursor by thymidine or dTMP, which provide a deoxyribose moiety, improves the growth of evolved D27F strains transformed with plasmid expressing active DeoB (*Figures 6B-C*). Moreover, to confirm that the route for deoxyribose degradation into energy production is blocked in evolved mutants, we measured the growth of naïve and evolved cells in the presence of thymidine as the sole carbon source and verified that *deoB*$^-$ cells cannot grow unless complemented with a plasmid expressing wild type DeoB (*Figure 6D*).

## Evolved D27 strains partially revert the omics effects of DHFR inactivation

Next, we carried out LC/MS TMT proteomics analysis of evolved strains (D27F, D27N and D27G) to help identify systems-level changes emerging from adaptation to lack of DHFR function. Since we observed that all clones randomly selected from trajectories that adapted to low folAmix concentrations showed very similar fitness profiles, we focused on individual clones arbitrarily chosen among D27F and D27N trajectories as representatives of adaptation to low folAmix. Specifically, clone F51T1#1 from D27F trajectory one and clone N51T6#1 from D27N trajectory six were selected. Likewise, clone G33T6#1 from D27G trajectory six was chosen as representative of adaptation through

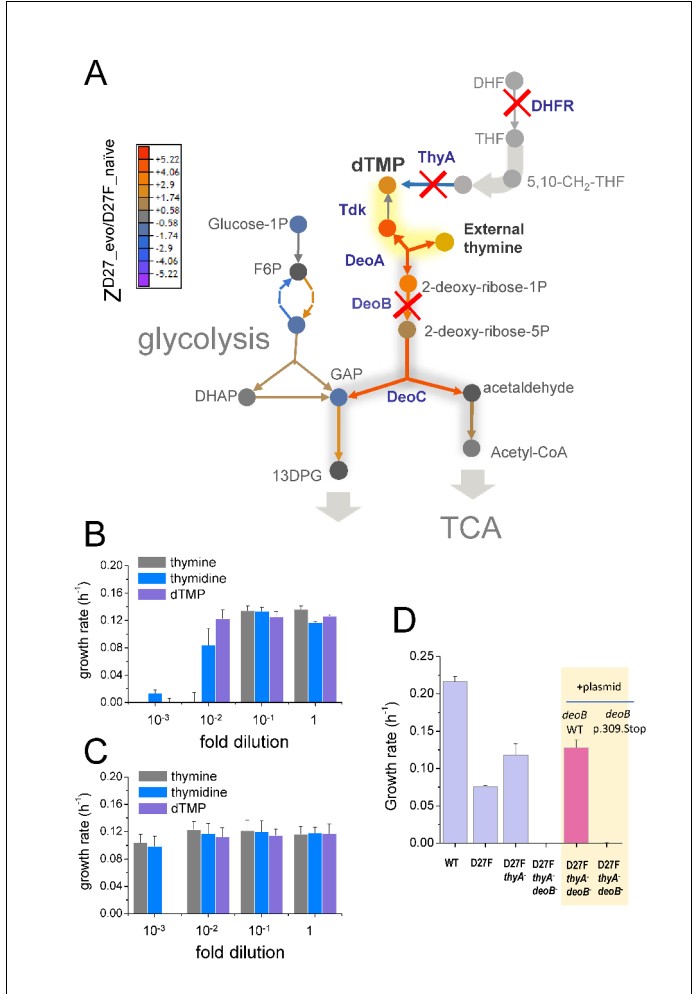

**Figure 6.** Inactivation in *deoB* gene prevents 2-deoxy-D-ribose-1-phosphate from being diverted into energy production via glycolysis and tricarboxylic acid cycle. **A)** Changes in the levels of metabolites (depicted as circles) and proteins (depicted as arrows) in the evolved D27F strain (Z-scores, relative to naïve D27F mutant) are represented. The proteins involved in 2-deoxy-D-ribose-1-phosphate degradation are marginally increased in evolved D27F, with respect to naïve strain (average Z-score = 0.445, p=0.0817, see *Figure 7—figure supplement 1*). However, deoB inactivation saves 2-deoxy-D-ribose-1-phospahte that is required in dTMP production, instead of being utilized for energy production in glycolysis, which is significantly upregulated in evolved D27F, with respect to naïve D27F (average Z-score = 0.593, p=0.015). **B)** Evolved D27F transformed with a plasmid expressing wild type DeoB are able to growth at low external concentrations of thymidine and dTMP, but not at low concentrations of thymine. **C)** Evolved D27F strains transformed a plasmid expressing inactive DeoB mutant grow at the same extent with either thymine, thymidine or dTMP. At the lowest concentration of dTMP, however, these cells are not able to grow, likely because of inefficient transport of this metabolite into the cell. **D)** DeoB⁻ strains cannot grow on thymidine as the sole carbon source unless complemented with a plasmid expressing wild type DeoB. Data are represented as mean ± SD (N = 3).

DOI: https://doi.org/10.7554/eLife.50509.020

The following source data is available for figure 6:

**Source data 1.** Inactivation in deoB gene prevents 2-deoxy-D-ribose-1-phosphate from being diverted into energy production.

DOI: https://doi.org/10.7554/eLife.50509.021

reversion of DHFR function. To get a glimpse of global proteomics changes we applied PCA analysis which revealed that both wild type and evolved D27G occupy the same quadrant in the space of two principal components, whereas D27 mutants that are folAmix dependent cluster more closely in another quadrant (*Figure 7—figure supplement 1A*). We then computed the Z-scores for the

proteomic changes in evolved strains with respect to naïve D27F ($Z^{D27\_evo/D27F\_naïve}$), to assess the effect of evolution on the proteomic levels and plotted these values against $Z^{D27F\_naïve/WT}$ to compare with the initial changes caused by D27F mutation (see also *Supplementary file 1* and *Supplementary file 2*). A clear anticorrelation was observed for all mutants (*Figure 7A*), being the strongest for evolved D27G strain.

These results indicate that adaptation leads to proteomic changes that generally oppose the immediate effects caused by DHFR inactivation. In the case of D27G, the strong global proteomic shift towards wild type levels is somewhat expected since in the evolved strain, the DHFR catalytic activity is partly restored by G27- > C mutation. To better characterize the proteomic changes that were specific to evolution of folAmix dependence we searched for classes of proteins grouped by function with significantly altered protein abundance levels in both evolved D27F and D27N with respect to naïve D27F strains (*Figure 7—figure supplement 1B*), as described previously. We found a significant increase in the expression levels of proteins involved in glycolysis, fermentation, sugar alcohol degradation and response to low pH, suggesting a metabolic shift towards mixed acid fermentation as a result of adaptation that blocked using derivative of thymidine for glycolysis to save them for DNA synthesis. On the other hand, the levels of proteins involved in DNA restriction/methylation and D-ribose uptake were significantly decreased with respect to naïve D27F in both evolved D27F and D27N strains. Next, we focused on evolved D27F strain to perform a detailed metabolomic analysis and characterize the changes occurring at the level of metabolites (see also *Supplementary file 3*). We computed Z-scores for metabolite changes with respect to naïve D27F ($Z^{D27\_evo/D27F\_naïve}$) and we found significant anticorrelation with Z-scores obtained for naïve D27F ($Z^{D27F\_naïve/WT}$) (*Figure 7B*), indicating that metabolite concentrations in evolved strain generally change to partially recover wild type levels, as observed with proteomics. We found that the pathways significantly enriched in metabolites with the highest $Z^{D27\_evo/D27F\_naïve}$ scores (>1.96) in evolved D27F were the same as those that had the most significant changes in naïve D27F, with respect to wild type (*Figure 2—figure supplement 4A–B*). Overall these results show that adaptation to low folAmix concentrations is accompanied by an overall shift in the abundance of metabolites and proteins to partially revert the system-level changes that are caused by DHFR inactivation.

## Growth inhibition by purine supplementation is suppressed in evolved D27F strain

Metabolites of purine and pyrimidine biosynthetic pathways were among the most strongly depleted in naïve D27F strain and showed highest increase upon evolution (*Figure 7C,D* and *Figure 2—figure supplement 4A–B*). We reasoned that these depleted metabolites could be limiting growth of the D27 mutants, and that supplementing the culture medium with these metabolites would improve growth. Surprisingly, supplementation with purines strongly *inhibited* growth of the naïve D27F strain, whereas pyrimidines addition had, at most, a slight stimulatory effect (*Figure 7E*). The detrimental effect of individual purine supplementation appears to be associated with regulatory responses and/or with toxic effects caused by metabolite imbalance (e.g. purines vs pyrimidines) upon supplementation. In the case of evolved D27F strain, the effect of supplementation of individual nucleotides on growth was comparably weaker or non-existent, indicating that the inhibitory effects were relaxed upon adaptation. Since we found no mutations directly associated with regulatory genes, it seems that the observed changes result solely from the re-wiring of metabolic pathways. In this regard our examples are particularly noteworthy. Evolution of D27F and D27N strains blocked supplementary glycolysis pathway by rerouting Deoxyribose towards DNA synthesis while the ensuing metabolic changes lead to significant increase in abundance of glycolysis proteins, presumably to compensate for diminished flux of sugar metabolites along the glycolysis pathway (see *Supplementary file 2*). Another example of metabolite-induced rewiring is purR operon whereby purR is inhibited by purine hypoxanthine. Evolved strains show significant increase in abundance of hypoxanthine giving rise to significant drop in abundance of proteins controlled by purR (see *Supplementary file 2*).

Overall, these results show that two key loss of function mutations that are responsible for the partial adaptation of D27F to loss of DHFR activity result in significant metabolic, proteomic and regulatory shifts observed in the evolved strains.

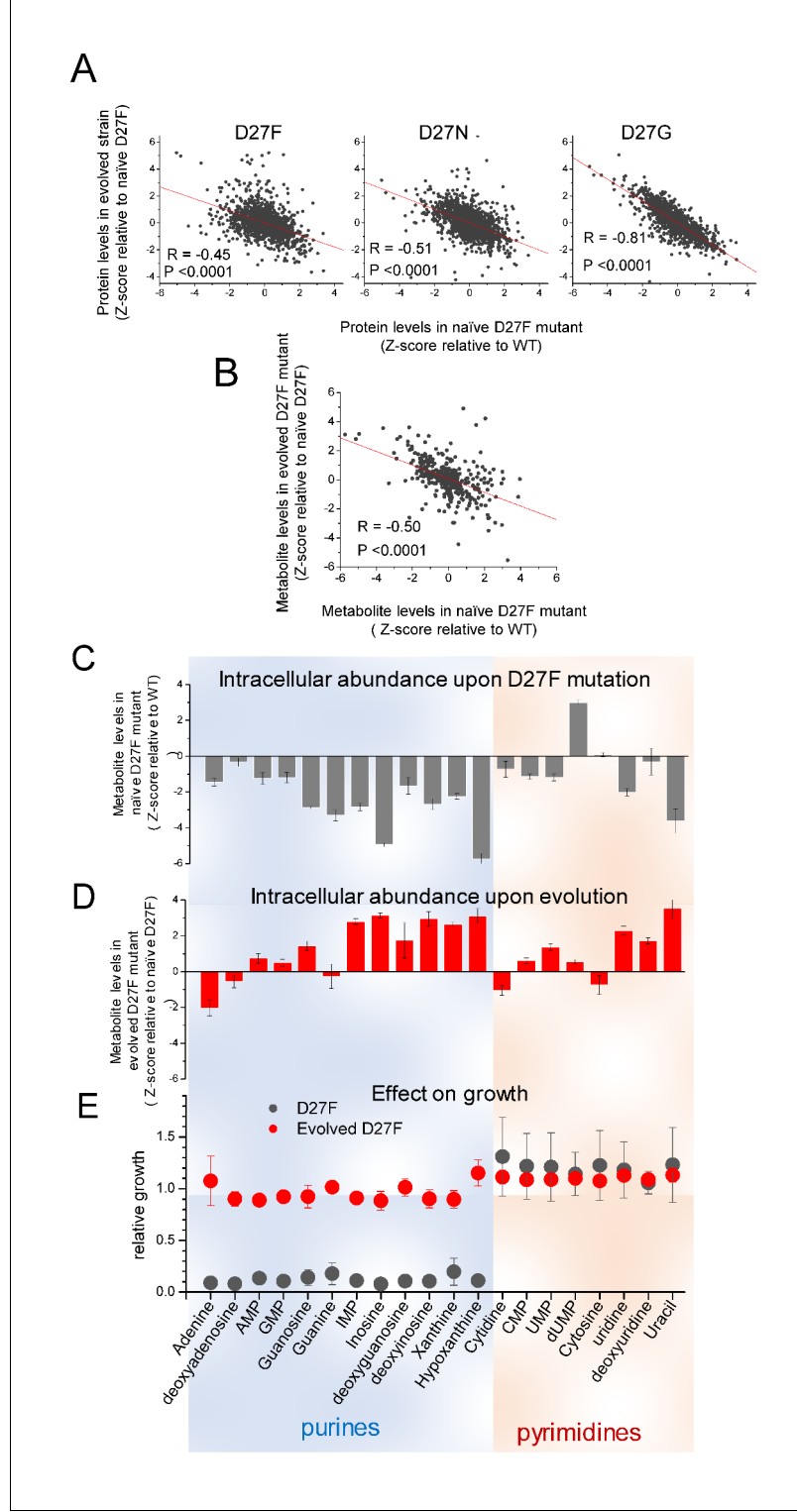

**Figure 7.** D27 mutation causes major metabolic and proteomic changes. **A**) Proteomics changes upon evolution partially revert the effect of DHFR inactivation (see also *Figure 2—figure supplement 4*). Comparison of the changes in protein levels obtained for evolved D27F (trajectory 1, cloneF51T1#1), D27N (trajectory 6, clone N51T6#1) and D27G (trajectory 6, clone G33T6#1) (Z-scores, relative to naïve D27F) with those measured for naïve D27F (Z-scores, relative to wild type). See also *Figure 7—figure supplement 1*. **B**) Comparison of the changes in metabolite levels measured for evolved D27F (trajectory 1, cloneF51T1#1) (Z-scores, relative to naïve D27F) with those measured for naïve D27F (Z-scores, relative with wild type). **C–E**) Inhibitory effect of purine metabolites

*Figure 7 continued on next page*

*Figure 7 continued*
supplementation on the growth of the naïve D27F strain is attenuated in the evolved D27F strain. **C**) Changes in intracellular levels of various purine and pyrimidine species determined by targeted metabolomics for naïve D27F strain, represented as Z-scores relative to wild type. **D**) Metabolite levels changes obtained for evolved D27F strain, represented as Z-scores relative to naïve D27F strain. **E**) Effect of individual metabolites on growth of naïve and evolved D27F mutant strains. The effect of supplementation with individual metabolites on growth was determined by growth measurements of naïve and evolved D27F mutant strains in M9 minimal medium supplemented by 1.6 mM thymine in combination with one of each tested metabolites (1 mM final concentration). Relative growth represents the maximum OD obtained for each metabolite normalized to the value measured in the presence of thymine alone. Although most purine metabolites are strongly depleted in naïve D27F mutant, supplementation of the culture medium with these compounds has a strong inhibitory effect on growth. Contrastingly, pyrimidines do not affect growth. In evolved D27F strain, the supplementation with purines has only a marginal effect on growth. Data are represented as mean ± SD (N = 3).
DOI: https://doi.org/10.7554/eLife.50509.022
The following source data and figure supplement are available for figure 7:

**Source data 1.** D27 mutation causes major metabolic and proteomic changes.
DOI: https://doi.org/10.7554/eLife.50509.024
**Figure supplement 1.** Proteomics profile of D27 mutants is associated with growth dependence on folAmix.
DOI: https://doi.org/10.7554/eLife.50509.023

## Discussion

In this study we set to explore the evolutionary mechanisms that follow inactivation of the essential gene that encodes DHFR. Such scenario is akin to antibiotic-mediated target inhibition, but distinct in that there is no selection for mutations that affect interactions with the drug, such as target modification and drug efflux mechanisms. For example, when treated with DHFR inhibitor trimethoprim, bacterial cells recurrently acquire resistance by high-fitness mutations near the active site of the target protein that decrease drug binding affinity, even at the expense of catalytic efficiency (*Rodrigues et al., 2016*; *Tamer et al., 2019*; *Toprak et al., 2012*). In contrast, a genetic perturbation in the same target gene provides the opportunity to study other potential unexplored mechanisms of adaptation in the absence of drug and evaluate their impact in the context of antibiotic resistance.

We found that, in the conditions studied here, evolution is constrained towards two solutions that appear to be mutually exclusive, either (partial) restoration of DHFR catalytic activity or adaptation to lack of DHFR function. While D27F, D27N and one trajectory in D27G convergently adapted to lack of DHFR function, other trajectories in D27G mutant reached reversion from thymine auxotroph phenotype. It is important to address the possible reasons for the observed outcome. A potentially influential factor could be the presence of background mutations observed in both D27N and D27G. The point mutation L44Q in *thyA*, reported to affect the activity of a nearly identical orthologue enzyme (*Nur-E-Kamal et al., 1994*), could have skewed the solution towards further inactivation of *thyA* by subsequent point mutations in other loci of that gene that were later observed in different trajectories. On the other hand, mutations in the genes *upp* and *ymfD* found in D27G strains did not impact the growth rate, thus the role of these mutations in altering the accessibility of the DHFR functional reversion G27- > C seems to be neutral.

The codon structure of each mutant studied here and bias in the frequency of each mutation type can also affect the likelihood that phenotype-reverting mutations may arise. We note that the nearest accessible high-fitness Asp/Glu codon for D27F mutant is two nucleotide substitutions away (although one nucleotide away from one D27C codon), whereas both D27G and D27N could be rescued by a single G->A or A->G transition, respectively. Only D27G was found to revert, however, not to wild type codon, but to a less catalytically efficient cysteine residue caused by a G->T transversion. In this respect, it is rather surprising that D27C mutation was fixed in three trajectories when transversions are generally regarded to be less likely than transitions. That this mutant was able to fix, despite its poor enzymatic activity, brings important insights on how deleterious mutations can be maintained when the selection pressure on protein function is alleviated by particular environmental conditions, in this case the presence of folAmix, and may provide a rapid route to the development of resistance. In addition, these results may hint at a possible mechanism for the divergence

of the key residues in the active sites of enzymes across long evolutionary timescales, where fixation of non-consensus mutations, such as D27C, may be allowed by transient adaptation to permissive environments, and subsequent compensatory mutations that restore protein function at later stages. While the selection pressure regime used in our study was solely based on the feedback loop strategy to control the concentration of folAmix, it remains to be explored how the implementation of alternative selection protocols could influence the course of evolution. Nonetheless, it appears that the system studied in this work provides an excellent framework for future theoretical and experimental studies to evaluate the role of environmental pressure dynamics in shaping evolution.

The most crucial factor in determining the fate of D27 mutant evolution appears to be the higher accessibility of mutational routes for inactivating *thyA* that lead to their fixation very early in the evolution experiment. Thymidylate synthase is the only known enzyme in *E. coli* able to synthetize dTMP de novo from dUMP, and inactivation of *thyA* inevitably commits the cell to thymine auxotrophy; reversion of DHFR function on the *thyA*⁻ background is not expected to change this phenotype, as it is known that *folA*⁺ *thyA*⁻ strains are thymine-dependent (**Bertino and Stacey, 1966**; **Schober et al., 2019**). Competition between DHFR reversion and *thyA* inactivation thus appears to be decisive in determining the evolutionary solution. However, since inactivation of a gene can be achieved by multiple ways, there is greater number of accessible evolutionary trajectories leading to *thyA* knockout than to reversion of DHFR function. This view has major implications for our understanding of how the relative impact of height and accessibility of fitness peaks shapes the evolutionary dynamics short-term adaptation.

Upon *thyA* inactivation, the only available route for dTMP synthesis involves DeoA-catalyzed formation of thymidine from thymine and 2-deoxy-D-ribose-1-phosphate (***Figure 6F***). However, DeoA is part of the *deo* operon that is induced by high levels of 2-deoxy-D-ribose-5-phosphate. This regulatory scheme allows cells to recycle excess metabolites into energy production. However, on the background of *thyA* inactivation, the co-expression of *deo* operon proteins creates simultaneously a solution and a problem for the uptake of thymine. Now DeoA becomes an essential protein, but its function is opposed by the roles of DeoC and DeoB which divert 2-Deoxy-D-ribose-1-phosphate into degradation via the glycolysis pathway. The evolutionary solution found in D27 strains converged to the inactivation of a single gene of the operon, *deoB*, an event that is sufficient to block the degradation pathway without disrupting the regulatory structure. This critical example illustrates how evolution can be constrained by regulatory circuits that provide conflicting responses when the optimal directionality of metabolic routes is switched due to genetic perturbations. Understanding these processes is critical for guidance towards new strategies for antibiotic resistance prevention. Perhaps not surprisingly, we found that pathways of adaptation to genetic inactivation of an antibiotic target provide a route to high levels of resistance. Likewise, a recent study also reported that *E. coli* cells challenged with increasing concentrations of trimethoprim in the presence of thymidine repeatedly evolved high levels of resistance by *thyA* loss-of-function mutations besides acquisition of commonly observed resistant mutations in DHFR (**Schober et al., 2019**). Strikingly, mutations in *thyA* are also known to confer trimethoprim resistance in bacterial clinical isolates of *S. aureus* (**Chatterjee et al., 2008**; **Kriegeskorte et al., 2014**), and *H. influenza* (**Rodríguez-Arce et al., 2017**), which emphasizes the need of laboratory studies of microbial adaptation as useful models to tackle serious global threat.

More generally this study highlights the evolutionary conundrum between 'survival of the fittest' and 'survival of the fastest'. Unlike previous studies that followed evolution of adaptation to gene knockouts, the current setup provides an 'easy' highest fitness solution by genetic reversion of single amino acid substitution. However, for most evolutionary trajectories the dynamics quickly converged to a 'quick and dirty' solution – a dead-end local fitness peak which provided an immediate relief from stress but blocked the path to highest fitness peak. Future studies will reveal how general this scenario is for evolutionary dynamics of adaptation to mutational inactivation and/or antibiotic induced inhibition of essential bacterial enzymes and beyond.

## Materials and methods

### Construction of plasmid pKD13-kefC(897–1863)-cmR-folA(wt)-kanR-apaH-apaG

A portion of chromosomal *apaH:apaG* genes was amplified by PCR using primers P4_chrom_flanking_for and PCRseq_apaH_rev. Then the pKD13 plasmid was linearized by PCR using the primers P4_chrom_flanking_for and PCRseq_apaH_rev. Both PCR products were purified and combined in a new PCR reaction using phosphorylated primers pKD13_post_downstream_for and PCRseq_apaH_rev. The linear product was ligated using T4 ligase to make plasmid Plasmid pKD13-*cmR*-folA(wt)-*kanR-apaH-apaG*. Then a portion of kefC gene was amplified by PCR using primers PCRseq_KefC_for2 and CapR-Chrom-Flanking rev. Then the plasmid pKD13-*cmR*-folA(wt)-*kanR-apaH-apaG* was linearized by PCR using primers Upstream_capR_for and pKD13_post_upstream_rev. Both products were combined in a new PCR reaction using phosphorylated primers PCRseq_KefC_for2 and pKD13_post_upstream_rev. Finally, the linear product was ligated using T4 ligase to make pKD13-*kefC*(897–1863)-*cmR*-folA(wt)-*kanR-apaH-apaG*.

### Mutations in D27 folA locus

Plasmid pKD13-*kefC*(897–1863)-*cmR*-folA(wt)-*kanR-apaH-apaG* was linearized using phosphorylated mutation primers D27Xmut_For and primer D27_rev, and ligated using T4 Ligase.

### Construction of D27 mutant strains

Inactive D27 mutants were created by lambda-red recombination (*Datsenko and Wanner, 2000*) according to a previously described procedure (*Bershtein et al., 2012*) with some modifications. Briefly, a pKD13 plasmid was modified to contain the entire regulatory and coding sequence of folA gene, flanked by two different antibiotic markers (genes encoding kanamycin (kanR) and chloramphenicol (cmR) resistances) and approximately 1 kb homologous region of both upstream and downstream chromosomal genes flanking folA gene (kefC and apaH, respectively). A list of relevant primers ais shown in *Supplementary file 6* . The entire cassette was amplified using primers PCRseq_KefC_for2 and PCRseq_apaH_rev and transformed into BW25113 cells with induced Red helper plasmid pKD46, and cells were recovered in SOC medium containing 1x folAmix (adenine 20 ug/mL, inosine 80 ug/mL, thymine 200 ug/mL, methionine 20 ug/mL and glycine 20 ug/mL). Transformants were plated in agar media containing both antibiotics and folAmix. To confirm correct integration of the desired mutations, folA locus was amplified by PCR (primers Ampl_RRfolA_for, Ampl_RRfolA_rev) and Sanger-sequenced using primer PCRseq_RRfolA_rev. Plasmid pKD46 was removed by plating cells at 37°C twice in the absence of antibiotic selection. Prior to starting the evolution experiments, the cultures had gone through three bottlenecks that involved randomly picking single colonies from agar plates for the subsequent step required for genetic manipulation.

### Automated experimental evolution

A liquid handling instrument Tecan Freedom Evo 150 equipped with Tecan Infinite M200 Pro plate reader and Liconic shaker was used in this work. The experiments were done at 30 °C and using M9 minimal media supplemented with 2 g/L glucose, 34 µg/mL chloramphenicol and 50 µg/mL kanamycin. The liquid handling worktable has two 100 mL troughs (Tecan) with either fresh M9 medium + antibiotics or 40% glycerol. In addition, solutions of folAmix at various concentrations (prepared in M9 + antibiotics) are provided in 50 mL falcon tubes. The general procedure involves up to four 96-well plates that are used for serial dilutions of bacterial cultures. Each working plate can carry eight trajectories that are positioned in a single column. In this work we used the first and eight wells in the column with media alone to control for contamination. The experiment starts by placing the 200 µL cultures/control in the first column of the 96-well plate. All the four plates are incubated in a shaker at 30 °C and at every 30 min the OD of each plate is determined alternately. The growth rate of each culture is calculated from OD measurements over time. To ensure proper comparison, the growth rate was computed only from OD values within a specified range (0.1–0.25). When the average OD of the six experimental replicates in a plate exceeds a threshold of 0.30 each culture is diluted into the next adjacent column by mixing a calculated volume of both culture and fresh medium and folAmix so that the initial OD is 0.01 in a total of 200 µL. At this point, the remaining

portion of the previous culture is mixed with glycerol in an auxiliary plate and subsequently frozen at −80 °C. The very first cultures to be frozen at −80 °C (at passage zero), are considered to be the naïve strains, and referred as such in the text to contrast with evolved strains which are picked at later passages. The cycle is repeated throughout the entire experiment. At every passage, the growth rate is compared with a threshold value (0.16 h$^{-1}$) and whenever a culture exceeds this value the concentration of folAmix is halved. The volume of folAmix added in each passage is 10, 5 or 2.5 µL.

## Media and growth conditions

Cell cultures were recovered by inoculating fresh M9 media medium supplemented with 0.2% glucose and 1x folAmix with a portion of −80 °C glycerol stocks taken at different passages. After recovery for several hours, the cultures were plated in agar media containing 1x folAmix, 34 µg/mL chloramphenicol and 50 µg/mL kanamycin and incubated at 30 °C. Plating evolved cultures at high folAmix concentrations ensures that the clones growing in agar media are not biased towards high fitness phenotypes. Individual colonies were randomly picked, grown in M9-media+folA mix and stored in glycerol at −80 °C for later analysis. Clones selected in this manner were labeled using the following mask: XdTt#*i*, where X refers to the amino acid letter in D27 locus (F/N/G), d is day at which the culture was passaged in the evolution experiment, t is the trajectory number (1-6) and *i* is the unique number of different colonies taken from the same culture. Cell cultures grown overnight were diluted in fresh M9 minimal media containing 1x folAmix and antibiotics and were grown for additional 4–6 hr. Cultures were then pelleted by centrifugation and washed three times in fresh M9 without folAmix. Microplates containing 150 µL of M9 minimal with 0.8 g/L glucose, antibiotics, and varying concentrations of folAmix were then inoculated with each culture at a starting OD of 0.0005. Growth measurements were performed in a Infinite 200 PRO plate reader for 48 hr at 30 °C with constant shaking. Growth rate values are represented as mean ± SD from at least three biological replicates.

## Protein purification and characterization

D27 mutant fused to C-terminal (6x) His-tag were overexpressed using pFLAG expression vector under isopropyl β-D-1-thiogalactopyranoside (IPTG) inducible T7 promoter. The recombinant proteins were purified from lysates on Ni-NTA columns (Qiagen) as described previously (*Rodrigues et al., 2016*).

## Determination of kinetic parameters

DHFR kinetic parameters were derived from the analysis of progress-curve kinetics of NADPH oxidation in the presence of different concentrations dihydrofolate using the software Kintek Explorer (*Johnson, 2009*) as described before (*Rodrigues et al., 2016*). The data correspond to the mean ± S.E. of at least three measurements.

## bis-ANS fluorescence measurements

DHFR protein solutions (2 µM) in the presence of 12 µM of bis-ANS were prepared in 50 mM phosphate and 1 mM DTT at pH 7.0 and placed in a 1 cm path-length quartz cuvette. The samples were equilibrated for 5 min at 37°C and the fluorescence emission spectra between 460 and 600 nm were recorded upon excitation at 395 nm. The emission band was integrated and the background of bis-ANS fluorescence in the absence of protein was subtracted. The area of the band was normalized to wild type *E. coli*. DHFR. The data corresponds to the mean ± S.E. of three measurements.

## Determination of the melting temperature

DHFR solutions (5 µM) were prepared in 50 mM phosphate buffer and 1 mM DTT at pH 7.0 in the presence of 100 µM NADPH. A temperature ramp of 1 °C/min was set between 25°C and 90°C, and the tryptophan fluorescence was recorded by measuring the intensity at 370 and 320 nm upon excitation at 280 nm. Thermal denaturation curves were also performed in the presence of 5x concentration of Sypro Orange and 20 µM. Melting temperature and confidence intervals at 95% were determined from the simultaneous analysis of the thermal denaturation curves obtained with both

tryptophan and Sypro Orange fluorescence using the online software Calfitter (*Mazurenko et al., 2018*).

## Measurements of total enzymatic activity in cell lysates

Cells from 14 mL cultures grown at 30°C at an OD of 0.5 were pelleted by centrifugation and lysed with 200 μL of lysis buffer containing 1 × Pop Culture reagent (Merck Millipore), 100 mM MES (2-(N-morpholino)ethanesulfonic acid) at pH 7.0 in the presence of 1 mM DTT in the presence of 1 × complete protease inhibitor mixture (Roche) and 1 mg/mL lysozyme and incubated for 20 min at room temperature. The lysate was sonicated with 10 pulses of 1 s,cleared by centrifugation and then 85 μL of the soluble fraction was transferred to 96-well white plates for total activity measurements using a total assay volume of 100 μL. The lysate was preincubated with 100 μM NADPH and the reaction was started with the addition of 100 μM dihydrofolate and mixing. The decrease in fluorescence (excitation at 300 nm and emission at 440 nm) was measured over 60 min at 25°C, and the initial velocities were computed. Measurements were also done in the presence of 1 mM TMP to control for the non-DHFR-specific reaction. Data corresponds to the mean ± S.E. of three measurements.

## Predicting fitness of D27 mutants

The fitness of *E. coli* DHFR mutants can be predicted from in vitro biophysical parameters using a simple metabolic model described in an earlier work (*Rodrigues et al., 2016*). The metabolic flux through DHFR reaction in DHFR mutant strains, normalized to wild type, can be approximated to:

$$V_{dhfr}^{norm} = \frac{\frac{k_{cat}^{mut}}{K_M^{mut}} \cdot [DHFR]^{mut}}{\frac{k_{cat}^{EcoliWT}}{K_M^{EcoliWT}} \cdot [DHFR]^{WT}} \cdot \frac{1}{\left(1 + \frac{\alpha \cdot [TMP]_{medium}}{K_i^{mut}}\right)} \tag{2}$$

Where, $\frac{k_{cat}}{K_M}$, $K_i$ and $[DHFR]$ are, respectively, the catalytic efficiency, trimethoprim inhibition constant and intracellular protein abundance of the mutant or of the wild type, and $[TMP]_{medium}$ and $\alpha$ are, respectively, the concentration of trimethoprim in the culture medium and the ratio of intracellular vs extracellular concentrations of trimethoprim, defined previously to be 0.1 (*Rodrigues et al., 2016*). To connect flux to fitness, we first measure the growth rate of wild type *E. coli* cells at various concentrations of DHFR inhibitor trimethoprim.

Then fitness is plotted against the flux through DHFR reaction calculated at each concentration of inhibitor using *Equation 2*, and catalytic parameters determined in vitro (*Table 1*). Protein abundance is determined by measuring the total catalytic activity in cell lysates, as described earlier (*Rodrigues et al., 2016*). Finally, we use *Equation 3*:

$$Gr^{norm} \sim \frac{V_{dhfr}^{norm}}{\left(B + V_{dhfr}^{norm}\right)} \tag{3}$$

to fit the fitness vs V$^{norm}$ data points to determine the parameter *B*, which represents the normalized flux at which growth rate is reduced by half. From this equation, the in vitro parameters determined for D27 mutants are used in *Equation 2 and 3* to predict fitness. The protein abundance of D27 mutants, however, cannot be determined from enzymatic assays in cell lysates, as these are inactive variants. Instead, protein abundance was predicted using an inverse relationship with relative bis-ANS fluorescence of those variants (*Bershtein et al., 2013*; *Rodrigues et al., 2016*). In qualitative terms, our model accurately predicted that the D27C variant is able to grow in the absence of folAmix supplement. However, the measured fitness was somewhat higher than the value predicted by the calculations. It is possible that some assumptions are not entirely met. One important assumption is that the concentration of DHFR and substrate dihydrofolate are either always constant or change only as a function of the flux calculated by *Equation 2*. The latter implies that, for every given value of flux, the changes in DHFR and dihydrofolate concentration will always be the same, and thus always comparable regardless of what parameters are used as input in *Equation 2*. However, this might not be always true. For example, a tight feedback control regulates the DHFR promoter activity, in response to the metabolic needs of the cell (*Bershtein et al., 2015a*; *Bershtein et al., 2015b*). Likewise, a decrease in DHFR catalytic function is also expected to result in

a considerable level of substrate build up. In these conditions, the magnitude of substrate $K_M$ for each variant might become more relevant. In our calculations, however, substrate saturation is not considered. This may lead to deviations, especially when KM values differ significantly, as we find in this work. How much these dynamic processes may differently affect the flux through DHFR reaction in each mutant is still to be determined.

### Effect of DeoB and ThyA expression

The genes *deoB* and *thyA* and corresponding mutants *deoB* p.309Stop and *thyA* 755-762del were cloned into a modified pTRC plasmid in which the *laqI* and promoter region were replaced by the *tetR* repressor gene and promotor region derived from plasmid Plasmid pEM-Cas9HF1-recA56 (addgene Addgene plasmid # 89962; *Moreb et al., 2017*). Transformed strain cells were grown in M9 minimal media + folAmix, then washed with fresh M9 media without folAmix and plated at different concentration of folAmix. Growth rate values are the mean ± SD of at least three biological replicates.

### Metabolite extraction and LC-MS analysis

Cultures of mutant strains (naïve and evolved D27F) and wild type (30 mL) were grown in parallel in M9 minimal media supplemented with 2 g/L glucose in a 250 mL flask at 30C. At an OD of 0.2–0.25 the cultures were centrifuged. The volume of culture to be pelleted was calculated so that the product OD ×volume of cell culture (mL) = 5. The pellet was mixed with 300 μl of 80:20 ratio of methanol:water that had been pre-chilled on dry ice. Samples were vortexed and incubated in dry ice for 10 min followed by centrifugation at 4 °C for 10 min at maximum speed. The supernatant was collected, and the pellet was processed by repeating the procedure. Samples were stored at −80 °C until analyzed by mass spectrometry. At least three independent biological replicates were analyzed for each strain. LC-MS analysis in the positive and negative mode was performed as previously described (*Bhattacharyya et al., 2016*). Retention times of several metabolites of the nucleotide biosynthesis pathway were determined from the analysis of pure compounds.

### LC/MS TMT Proteomics

Cell cultures from isolated colonies were grown at 30 °C in M9 minimal media supplemented with 0.2% glucose and were collected by centrifugation during mid-exponential phase (OD ~0.2–0.25), well below saturation due to oxygen limitation (typically at OD >0.8–0.9). Global proteomic analysis was performed as described previously (*Bershtein et al., 2015a*).

### Whole-genome sequencing

Sequencing was performed on isolated colonies on Illumina MiSeq in 2 × 150 bp paired-end configuration (Genewiz, Inc, South Plainfield, NJ). The raw data were processed with the with the breseq pipeline (*Deatherage et al., 2014*) on default settings, using the *E. coli* K-12 substr. BW25113 reference genome (GenBank accession no. CP009273.1). The experiment accession numbers for the raw reads deposit are as follows: Naïve D27F: SRX6873308, evolved F15T2#1: SRX6873309, evolved F22T2#1: SRX6873310, evolved F27T2#1: SRX6873311, evolved F32T2#1: SRX6873312, evolved F51T1#1: SRX6873313, naïve D27N: SRX6873314, evolved N51T6#1: SRX6873315, naïve D27G: SRX6873316, Evolved G33T6#1: SRX6873317. The BioProject accession number for the whole project is PRJNA566398.

### Metabolomics data analysis

Data analysis was performed using the software packages MzMatch (*Scheltema et al., 2011*) and IDEOM (*Creek et al., 2012*) for untargeted analysis. For peak assignment in untargeted analysis, IDEOM includes both peak m/z values and predicted retention times calculated based on chemical descriptors (*Creek et al., 2011*). A list of 32 experimentally measured retention times was initially used to calibrate retention time predictions. The retention time of putatively identified metabolites were found to correlate fairly well with the values included in IDEOM ($R^2$ = 0.73) and published in other studies (*Pluskal et al., 2010*) ($R^2$ = 0.88 and $R^2$ = 0.61, respectively)ee. For this reason, additional metabolites from those sources that closely matched IDEOM assignments were treated as standards in the identification routine. Following identification, Z-scores with respect to wild type

($Z^{D27F\_naïve/WT}$) or to naïve D27F strain ($Z^{D27F\_evo/D27F\_ naïve}$) were calculated using *Equation 1*. Z-scores determined for each set $i$ of $n$ biological replicates were combined for each metabolite *met*

using the expression $z_{combined}^{met} = \frac{\sum_i^n z_i^{met}}{\sqrt{n}}$. The set of metabolites with highest absolute combined Z-scores (>1.96) was selected to perform an overrepresentation (enrichment) analysis of categorical annotations using MBrole2.0 (*López-Ibáñez et al., 2016*), using as reference the metabolic pathways of *E. coli* from KEGG database.

## Proteomics data analysis

Z-scores were computed using *Equation 1*. For grouping analysis, we used the functional and regulatory classification group sets as (*Sangurdekar et al., 2011*). For each set of genes belonging to a group we employed a one-sample t-test which provides the p-value against the null hypothesis that the group of genes was drawn from a normal distribution and considered that a given group of genes is upregulated or downregulated with respect to a reference if the average of Z-scores of that group is positive or negative, respectively.

## Acknowledgements

This research was supported by NIH grant 5R01GM068670 to E I Shakhnovich.

## Additional information

### Funding

| Funder | Grant reference number | Author |
| --- | --- | --- |
| National Institute of General Medical Sciences | 068670 | João V Rodrigues Eugene I Shakhnovich |

The funders had no role in study design, data collection and interpretation, or the decision to submit the work for publication.

### Author contributions

João V Rodrigues, Conceptualization, Formal analysis, Validation, Investigation, Writing—original draft; Eugene I Shakhnovich, Conceptualization, Formal analysis, Supervision, Funding acquisition, Investigation, Writing—original draft, Writing—review and editing

### Author ORCIDs

João V Rodrigues (iD) https://orcid.org/0000-0002-5605-656X
Eugene I Shakhnovich (iD) https://orcid.org/0000-0002-4769-2265

### Decision letter and Author response

Decision letter https://doi.org/10.7554/eLife.50509.036
Author response https://doi.org/10.7554/eLife.50509.037

## Additional files

### Supplementary files

• Source code 1. Evolution script.
DOI: https://doi.org/10.7554/eLife.50509.025

• Supplementary file 1. TMT proteomics of naïve and evolved D27 strains. All abundances are measured relative to WT strain which is taken as reference. Z-scores of abundance variation are calculated as outlined in text (see also *Bershtein et al., 2015a*)
DOI: https://doi.org/10.7554/eLife.50509.026

• Supplementary file 2. Variation of abundances of proteins grouped into functional groups according to *Sangurdekar et al. (2011)*. All protein abundances are relative to WT obtained from TMT

proteomics (see *Supplementary file 1* for raw data). Average abundance variation z-score of proteins in the group and associated p-values are calculated as described in detail in *Bershtein et al. (2015a)*. Groups marked in red represent functions that exhibit statistically significant increase or decrease of abundances in evolved but not naïve D27 strains.

DOI: https://doi.org/10.7554/eLife.50509.027

• Supplementary file 3. Variation of abundances of metabolites in naïve and evolved D27 strains.
DOI: https://doi.org/10.7554/eLife.50509.028

• Supplementary file 4. WGS and Sanger sequencing data for various evolutionary trajectories in naïve and evolved D27 strains.
DOI: https://doi.org/10.7554/eLife.50509.029

• Supplementary file 5. Single point mutations in thyA and deoB are predicted to impair catalytic activity and/or destabilize the protein.
DOI: https://doi.org/10.7554/eLife.50509.030

• Supplementary file 6. Key Resource table.
DOI: https://doi.org/10.7554/eLife.50509.031

• Transparent reporting form  DOI: https://doi.org/10.7554/eLife.50509.032

### Data availability

All data are presented in files: Supplementary file 1, Supplementary file 2, Supplementary file 3, Supplementary file 4, Figure 2—source data 1, Figure 3—source data 1, Figure 4—source data 1, Figure 5—source data 1, Figure 6—source data 1, Figure 7—source data 1.

The following dataset was generated:

| Author(s) | Year | Dataset title | Dataset URL | Database and Identifier |
| --- | --- | --- | --- | --- |
| Rodrigues JR, Shakhnovich ES | 2019 | Resequencing evolved folA D27 mutant strains | https://www.ncbi.nlm.nih.gov/bioproject/?term=PRJNA566398 | NCBI Bioproject, PRJNA566398 |

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
