## [Decision Letter]

Thank you for submitting your article "Adaptation to mutational inactivation of an essential *E. coli* gene converges to an accessible suboptimal fitness peak" for consideration by *eLife*. Your article has been reviewed by three peer reviewers, one of whom is a member of our Board of Reviewing Editors, and the evaluation has been overseen by Michael Eisen as the Senior Editor. The following individual involved in review of your submission has agreed to reveal their identity: Erdal Toprak (Reviewer #2).

The reviewers have discussed the reviews with one another and the Reviewing Editor has drafted this decision to help you prepare a revised submission.

Summary:

In this study, the authors investigate the evolutionary pathways and phenotypes derived from inactivation of an essential enzyme (DHFR) in the *E. coli* bacterium with the objective of determining how adaptation occurs in bacterial systems and its mechanisms. Since the authors' approach does not involve antibiotic resistance or gene knockouts but compensation for loss of function mutations of an essential gene, the results provide a unique view into less explored mechanisms of adaptation. The authors discovered that there are two dominant pathways for adaptation, one that restores (albeit partially and through a different route) original functionality and a second that compensates the metabolic effects of the loss of function via the deactivation of other genes involved in metabolism. The authors conclude that given that there are many more pathways to disrupt other functions that will compensate the effects versus the unique pathway to completely revert the mutations in D27 of DHFR, the cell effectively converges to a local fitness peak as opposed to the global one. Thus, the key finding of the study is that the evolutionary pathways to adaptation were suboptimal (i.e., the organisms did not escape from supplement auxotrophy), largely because the adaptive mutations that were easiest to acquire were also the ones that blocked the path to the highest fitness peak.

This work is extensive both in depth and breadth. The authors utilize a large set of experimental tools ranging from biophysical measurements, whole genome sequencing, NGS, metabolomics and automated plating to not only support their message but to disentangle a series of mechanistic questions. All key questions are answered with solid and convincing experiments and this study constitutes a valuable example of how we can study evolutionary landscapes in a comprehensive way.

Essential revisions:

1) The manuscript was overall relatively easy to follow and the experiments well described and performed. We had trouble following the authors in some places, and arguably the reviewers' biggest concern is that the manuscript isn't as accessible to the general reader of *eLife* as it should be.

2) Along these lines, the beginning of the Discussion section explains very nicely the difference between the authors' study of adaptation that starts from a non-functional essential gene and classical studies that use drugs to generate essentiality. Given that this is a theme that the authors have done experiments on (with the use of the DHFR inhibitor trimethoprim), it would be good if the authors introduced the topic in the Introduction so that it's easier for readers to follow.

3) It was not clear to the reviewers how the authors selected mutations ad D27. The rationale is that mutations G,N and F would disrupt function without disrupting folding, but how did the authors narrow the possibilities to 3 amino acids?

4) The authors propose a model to predict the effects of the D27C mutation on catalytic activity as a way to support the notion that such mutant might be partially functional. Then they purified the mutant and assayed the DHFR activity. Although it seems nice that the model agrees with the experiment, it seems to the reviewers that the model is not needed any more and its relevance is diminished after the experimental confirmation.

5) It was not clear why if the D27C mutation yields low levels of activity the mutant restored its function without any other change. Is this low activity above the survival limit?

6) The authors note that the D27N and D27G strains had additional background mutations that are probably acquired prior to the evolution experiments. Could the authors clarify how do they know these mutations appeared before? Did they perform sequencing experiments prior to the evolution experiments? If so, they should mention it; if not, then an argument of why these mutations were acquired previously must be presented.

7) The section "Regulatory responses are altered in evolved D27F strain" seems more speculative and in our opinion is not as solid as the previous sections. Our suggestion is to move that section to the supplementary information. This would help streamline the article, which is already extensive, and will not change the main message and results of the article but could help improve the flow.

8) The feedback loop strategy seems really interesting and allows for the interesting observations in this work. Can the authors speculate (or comment, if they have done it) on what would happen if they try to evolve the strains without the feedback loop?

---

## [Author Response]

Essential revisions:1) The manuscript was overall relatively easy to follow and the experiments well described and performed. We had trouble following the authors in some places, and arguably the reviewers' biggest concern is that the manuscript isn't as accessible to the general reader of eLife as it should be.

We thank the reviewers for their helpful comments and constructive criticisms. We have implemented their suggestions, clarified some unclear aspects in the previous manuscript and provided a response to the raised questions. We believe that the changes made in the revised manuscript made it clearer and more accessible to the general *eLife* readership.

2) Along these lines, the beginning of the Discussion section explains very nicely the difference between the authors' study of adaptation that starts from a non-functional essential gene and classical studies that use drugs to generate essentiality. Given that this is a theme that the authors have done experiments on (with the use of the DHFR inhibitor trimethoprim), it would be good if the authors introduced the topic in the Introduction so that it's easier for readers to follow.

Following the reviewer’s suggestion, we added a sentence in the Introduction to introduce the topic of resistance to trimethoprim:

“Here we study evolutionary adaptation upon functional inactivation of an essential *E. coli* enzyme dihydrofolate reductase (DHFR). […] Past efforts to link fitness effects of chromosomal variation in the folA locus encoding DHFR and their biophysical effects on DHFR protein allowed us to develop an accurate quantitative biophysical model of DHFR fitness landscape (Bershtein et al., 2015; Bershtein et al., 2013; Bershtein, Mu, and Shakhnovich, 2012; Bershtein et al., 2015; Rodrigues et al., 2016).”

3) It was not clear to the reviewers how the authors selected mutations ad D27. The rationale is that mutations G,N and F would disrupt function without disrupting folding, but how did the authors narrow the possibilities to 3 amino acids?

The mutations introduced at position 27 were selected to disrupt DHFR activity and to change the local structure at the active site pocket. We wanted to explore if distinct local structural perturbations caused by replacing D27 by different residues could lead to alternative mutational pathways to restore DHFR function during evolution. To clarify the rationale for the choice of mutations we included the following sentence in the Results (subsection “D27 mutations confer growth defects”):

“These replacements were chosen to explore how different degrees of structural perturbations could potentially lead to alternative mutational pathways; while asparagine is structurally similar to aspartate, the introduction of residues with side chains that are either bulky and hydrophobic (phenylalanine) or absent (glycine) is expected to create significantly more structural perturbations, at least locally at the active site.”

4) The authors propose a model to predict the effects of the D27C mutation on catalytic activity as a way to support the notion that such mutant might be partially functional. Then they purified the mutant and assayed the DHFR activity. Although it seems nice that the model agrees with the experiment, it seems to the reviewers that the model is not needed any more and its relevance is diminished after the experimental confirmation.

We agree with the reviewers that predictions of fitness become less relevant once experimental data is available. However, we think that a graphical representation of the fitness landscape defined by the biophysical model can be useful so that readers can visually follow how mutations are positioned along that fitness landscape, especially to better appreciate the location of starting (D27G) and ending (D27C) evolutionary points on the fitness landscape. In addition, since the prediction made by the model doesn’t entirely quantitatively match the experimental value, we think it is appropriate to show this discrepancy and discuss the possible reasons for it, which should motivate further work to address this interesting issue

5) It was not clear why if the D27C mutation yields low levels of activity the mutant restored its function without any other change. Is this low activity above the survival limit?

It is not known why the cells tolerate such low DHFR catalytic activity nor what cellular mechanisms determine the survival limit in terms of DHFR function. However, the survival limit can be experimentally determined by growth measurements performed in the presence of different trimethoprim concentrations as discussed in our previous paper (Rodrigues et al., 2016). These measurements allow to calibrate the biophysical model of DHFR fitness to account for the survival limit threshold (determination of parameter B in Equation 1 of Rodrigues et al., (2016) by fitting growth data, as shown in Figure 3A of that paper). Indeed, the model predicts that the levels of catalytic activity in D27C mutant are above the survival limit.

6) The authors note that the D27N and D27G strains had additional background mutations that are probably acquired prior to the evolution experiments. Could the authors clarify how do they know these mutations appeared before? Did they perform sequencing experiments prior to the evolution experiments? If so, they should mention it; if not, then an argument of why these mutations were acquired previously must be presented.

The genetic manipulation required to introduce D27 mutations starting from the same wild type parent *E. coli* strain involves plating liquid cultures in agar plates and randomly picking single colonies for the subsequent steps. This means that any mutation randomly selected in these single cell bottlenecks will be present in the strain used in the evolution experiments.

We have clarified this in the Results section (subsection “Loss of function mutations in *thyA* and *deoB* lead to adaptation to low folAmix concentrations”) as follows:

“[…] although all D27 strains were constructed from the same wild type *E. coli* parent strain, we could not prevent the appearance of mutations at any given stage of genetic manipulation prior to the evolution experiments.”

This is also explained in the Materials and Methods subsection “Construction of D27 mutant strains”:

“Prior to starting the evolution experiments, the cultures had gone through 3 bottlenecks that involved randomly picking single colonies from agar plates for the subsequent step required for genetic manipulation.”

Analysis of naïve strains by next generation sequencing was done only after the evolution experiments had been completed, and thus the experiments were done without prior knowledge of the presence of additional background mutations.

In the second paragraph of Discussion section we also discuss how the presence of additional mutations could have affected the results:

“A potentially influential factor could be the presence of background mutations observed in both D27N and D27G. […] On the other hand, mutations in the genes *upp* and *ymfD* found in D27G strains did not impact the growth rate, thus the role of these mutations in altering the accessibility of the DHFR functional reversion G27->C seems to be neutral.”

7) The section "Regulatory responses are altered in evolved D27F strain" seems more speculative and in our opinion is not as solid as the previous sections. Our suggestion is to move that section to the supplementary information. This would help streamline the article, which is already extensive, and will not change the main message and results of the article but could help improve the flow.

We think the reviewer’s criticism about the section “Regulatory responses are altered in evolved D27F strain” being more speculative is valid. On the other hand, although the mechanistic explanation for the observations made was not pursued in detail in this work, we think that the results described in that section, namely the growth inhibition by external purines, are interesting for two reasons. First, the inhibitory effects of supplemented purines are counterintuitive since the intracellular abundance of these metabolites is extremely depleted in D27F strain. Second, the dramatic inhibitory effect is abolished in the evolved strains due to the acquisition of only a few key mutations. We think it is pertinent to discuss these results in the main text, together with metabolic and proteomic characterization, to provide a global picture of the systems-wide domino effect caused by D27 mutations and subsequent adaptation by loss-of-function mutations in 2 metabolic genes. To address the reviewer’s criticism, we have significantly reduced the size of the section in question and removed sentences that were speculative. The title of that section was also changed to “Growth inhibition by purine supplementation is suppressed in evolved D27F strain” since we cannot fully exclude other explanations besides an alteration in the regulatory responses.

8) The feedback loop strategy seems really interesting and allows for the interesting observations in this work. Can the authors speculate (or comment, if they have done it) on what would happen if they try to evolve the strains without the feedback loop?

The reviewer raises the important question of how different protocols for setting the selection pressure during the evolution experiments could influence the results. This is certainly an important direction for future studies, and we have included a sentence to the Discussion section to emphasize that point:

“While the selection pressure regime used in our study was solely based on the feedback loop strategy to control the concentration of folAmix, it remains to be explored how the implementation of alternative selection protocols could influence the course of evolution. Nonetheless, it appears that the system studied in this work provides an excellent framework for future theoretical and experimental studies to evaluate the role of environmental pressure dynamics in shaping evolution.”